# The power of absolute discounting: all-dimensional distribution estimation

**Moein Falahatgar**
UCSD
moein@ucsd.edu

**Mesrob Ohannessian**
TTIC
mesrob@gmail.com

**Alon Orlitsky**
UCSD
alon@ucsd.edu

**Venkatadheeraj Pichapati**
UCSD
dheerajpv7@ucsd.edu

## Abstract

Categorical models are a natural fit for many problems. When learning the distribution of categories from samples, high-dimensionality may dilute the data. Minimax optimality is too pessimistic to remedy this issue. A serendipitously discovered estimator, absolute discounting, corrects empirical frequencies by subtracting a constant from observed categories, which it then redistributes among the unobserved. It outperforms classical estimators empirically, and has been used extensively in natural language modeling. In this paper, we rigorously explain the prowess of this estimator using less pessimistic notions. We show that (1) absolute discounting recovers classical minimax KL-risk rates, (2) it is *adaptive* to an effective dimension rather than the true dimension, (3) it is strongly related to the Good–Turing estimator and inherits its *competitive* properties. We use power-law distributions as the cornerstone of these results. We validate the theory via synthetic data and an application to the Global Terrorism Database.

## 1 Introduction

Many natural problems involve uncertainties about categorical objects. When modeling language, we reason about words, meanings, and queries. When inferring about mutations, we manipulate genes, SNPs, and phenotypes. It is sometimes possible to embed these discrete objects into continuous spaces, which allows us to use the arsenal of the latest machine learning tools that often (though admittedly not always) need numerically meaningful data. But why not operate in the discrete space directly? One of the main obstacles to this is the dilution of data due to the high-dimensional aspect of the problem, where dimension in this case refers to the number $k$ of categories.

The classical framework of categorical distribution estimation, studied at length by the information theory community, involves a fixed small $k$, [BS04]. Add-contant estimators are sufficient for this purpose. Some of the impetus to understanding the large $k$ regime came from the neuroscience world, [Pan04]. But this extended the pessimistic worst-case perspective of the earlier framework, resulting in guarantees that left a lot to be desired. This is because high-dimension often also comes with additional structure. In particular, if a distribution produces only roughly $d$ distinct categories in a sample of size $n$, then we ought to think of $d$ (and not $k$) as the *effective* dimension of the problem. There are also some ubiquitous structures, like power-law distributions. Natural language is a flagship example of this, which was observed as early as by Zipf in [Zip35]. Species and genera, rainfall, terror incidents, to mention just a few all obey power-laws [SLE$^+$03, CSN09, ADW13].

Are there estimators that mold to both dimension *and* structure? It turns out we don't need to search far. In natural language processing (NLP) it was first discovered that an estimator proposed by Good and Turing worked very well [Goo53]. Only recently did we start understanding why and how [OSZ03, OD12, AJOS13, OS15]. And the best explanation thus far is that it implicitly *competes* with the best estimator in a very small neighborhood of the true distribution. But NLP researchers [NEK94, KN95, CG96] have long realized that another simpler estimator, *absolute discounting*, is equally good. But why and how this is the case was never properly determined, save some mention in [OD12] and in [FNT16], where the focus is primarily on form.

In this paper, we first show that absolute discounting, defined in Section 3, recovers pessimistic minimax optimality in both the low- and high-dimensional regimes. This is an immediate consequence of an upper bound that we provide in Section 5. We then study lower bounds with classes defined by the number of distinct categories $d$ and also power-law structure in Section 6. This reveals that absolute discounting in fact *adapts* to the family of these classes. We further unravel the relationship of absolute discounting with the Good–Turing estimator, for power-law distributions. Interestingly, this leads to a further refinement of this estimator's performance in terms of *competitivity*. Lastly, we give some synthetic experiments in Section 8 and then explore forecasting global terror incidents on real data [LDMN16], which showcases very well the "all-dimensional" learning power of absolute discounting. These contributions are summarized in more detail in Section 4. We start out in Section 2 with laying out what we mean by these notions of optimality.

## 2  Optimal distribution learning

In this section we concretely formulate the optimal distribution learning framework and take the opportunity to point out related work.

**Problem setting**   Let $p = (p_1, p_2, \ldots, p_k)$ be a distribution over $[k] := \{1, 2, \ldots, k\}$ categories. Let $[k]^*$ be the set of finite sequences over $[k]$. An estimator $q$ is a mapping that assigns to every sequence $x^n \in [k]^*$ a distribution $q(x^n)$ over $[k]$. We model $p$ as being the underlying distribution over the categories. We have access to data consisting of $n$ samples $X^n = X_1, X_2, ..., X_n$ generated *i.i.d.* from $p$. Intuitively, our goal is to find a choice of $q$ that is guaranteed to be as close as any other estimator can be to $p$, in average. We first need to quantify how performance is measured.

*General notation:* Let $(\mu_j \; : \; j = 1, \cdots, k)$ denote the empirical counts, i.e. the number of times symbol $j$ appears in $X^n$ and let $D$ be the number of *distinct* categories appearing in $X^n$, i.e. $D = \sum_j \mathbb{1}\{\mu_j > 0\}$. We denote by $d := \mathbb{E}[D]$ its expectation. Let $(\Phi_\mu \; : \; \mu = 0, \cdots, n)$, be the total number of categories appearing exactly $\mu$ times, $\Phi_\mu := \sum_j \mathbb{1}\{\mu_j = \mu\}$. Note that $D = \sum_{\mu>0} \Phi_\mu$. Also let $(S_\mu \; : \; \mu = 0, \cdots, n)$, be the total probability within each such group, $S_\mu := \sum_j p_j \mathbb{1}\{\mu_j = \mu\}$. Lastly, denote the empirical distribution by $q_j^{+0} := \mu_j/n$.

**KL-Risk**   We adopt the Kullback-Leibler (KL) divergence as a measure of loss between two distributions. When a distribution $p$ is approximated by another $q$, the KL divergence is given by $\mathsf{KL}(p||q) := \sum_{j=1}^k p_j \log \frac{p_j}{q_j}$. We can then measure the performance of an estimator $q$ that depends on data in terms of the *KL-risk*, the expectation of the divergence with respect to the samples. We use the following notation to express the KL-risk of $q$ after observing $n$ samples $X^n$:

$$r_n(p, q) := \mathop{\mathbb{E}}_{X^n \sim p^n}[\mathsf{KL}(p||q(X^n))].$$

An estimator that is identical to $p$ regardless of the data is unbeatable, since $r_n(p, q) = 0$. Therefore it is important to model our ignorance of $p$ and gauge the optimality of an estimator $q$ accordingly. This can be done in various ways. We elaborate the three most relevant such perspectives: *minimax*, *adaptive*, and *competitive* distribution learning.

**Minimax**   In the *minimax* setting, $p$ is only known to belong to some class of distributions $\mathcal{P}$, but we don't know which one. We would like to perform well, no matter which distribution it is. To each $q$ corresponds a distribution $p \in \mathcal{P}$ (assuming the class is finite or closed) on which $q$ has its *worst* performance:

$$r_n(\mathcal{P}, q) := \max_{p \in \mathcal{P}} r_n(p, q).$$

The minimax risk is the *least* worst-case KL-risk achieved by *any* estimator $q$,

$$r_n(\mathcal{P}) := \min_q r_n(\mathcal{P}, q).$$

The minimax risk depends only on the class $\mathcal{P}$. It is a *lower bound*: no estimator can beat it *for all $p$*, i.e. it's not possible that $r_n(p, q) < r_n(\mathcal{P})$ for all $p \in \mathcal{P}$. An estimator $q$ that satisfies an *upper bound* of the form $r_n(\mathcal{P}, q) = (1 + o(1))r_n(\mathcal{P})$ is said to be minimax *optimal* "even to the constant" (an informal but informative expression that we adopt in this paper). If instead $r_n(\mathcal{P}, q) = \mathcal{O}(1)r_n(\mathcal{P})$, we say that $q$ is *rate optimal*. Near-optimality notions are also possible, but we don't dwell on them. As an aside, note that *universal compression* is minimax optimality using *cumulative* risk. See [FJO$^+$15] for such related work on universal compression for power laws.

**Adaptive**   The minimax perspective captures our ignorance of $p$ in a pessimistic fashion. This is because $r_n(\mathcal{P})$ may be large, but for a specific $p \in \mathcal{P}$ we may have a much smaller $r_n(p, q)$. How can we go beyond this pessimism? Observe that when a class is smaller, then $r_n(\mathcal{P})$ is smaller. This is because we'd be maximizing on a smaller set. In the extreme case noted earlier, when $\mathcal{P}$ contains only a single distribution, we have $r_n(\mathcal{P}) = 0$. The *adaptive* learning setting finds an intermediate ground where we have a *family* of distribution classes $\mathcal{F} = \{\mathcal{P}_s : s \in \mathcal{S}\}$ indexed by a (not necessarily countable) index set $\mathcal{S}$. For each $s$, we have a corresponding $r_n(\mathcal{P}_s)$ which is often much smaller than $r_n \left( \bigcup_{s \in \mathcal{S}} \mathcal{P}_s \right)$, and we would like the estimator to achieve the risk bound corresponding to the smaller class. We say that an estimator $q$ is *adaptive* to the family $\mathcal{F}$ if for all $s \in \mathcal{S}$:

$$r_n(p, q) \leq O_s(1)\, r_n(\mathcal{P}_s) \quad \forall p \in \mathcal{P}_s \quad \Longleftrightarrow \quad r_n(\mathcal{P}_s, q) \leq O_s(1)\, r_n(\mathcal{P}_s)$$

There often is a price to adaptivity, which is a function of the granularity of $\mathcal{F}$ and is paid in the form of varying/large leading constants per class. This framework has been particularly successful in density estimation with smoothness classes [Tsy09] and has been recently used in the discrete setting for universal compression [BGO15].

**Competitive**   The adaptive perspective can be tightened by demanding that, rather than a multiplicative constant, the KL-risk tracks the risk up to a vanishingly small *additive* term:

$$r_n(p, q) = r_n(\mathcal{P}_s) + \epsilon_n(\mathcal{P}_s, q) \quad \forall p \in \mathcal{P}_s.$$

Ideally, we would like the *competitive loss* $\epsilon_n(\mathcal{P}_s, q)$ to be negligible compared to the risk of each class $r_n(\mathcal{P}_s)$. If $\epsilon_n(\mathcal{P}_s, q) = O_s(1)r_n(\mathcal{P}_s)$ for all $s$, then we recover adaptivity. And when $\epsilon_n(\mathcal{P}_s, q) = o_s(1)r_n(\mathcal{P}_s)$ for all $s \in \mathcal{S}$, we have minimax optimality even to the constant within each class, which is a much stronger form of adaptivity. We then say that the estimator is *competitive* with respect to the family $\mathcal{F}$. We may also evaluate the *worst-case* competitive loss, over $\mathcal{S}$.

This formulation was recently introduced in [OS15] in the context of distribution learning. This work shows that the celebrated Good–Turing estimator [Goo53], combined with the empirical estimator, has small worst-case competitive loss over the family of classes defined by any given distribution and all its permutations. Most importantly, this loss was shown to stay bounded, even as the dimension increases. This provided a rigorous theoretical explanation for the performance of the Good–Turing estimator in high-dimensions. A similar framework is also studied for $\ell_1$-loss in [VV15].

## 3   Absolute discounting

One of the first things to observe is that the empirical distribution is particularly ill-suited to handle KL-risk. This is most easily seen by the fact that we'd have infinite blow-up when any $\mu_j = 0$, which *will* happen with positive probability. Instead, one could resort to an add-constant estimator, which for a positive $\beta$ is of the form $q_j^{+\beta} := (\mu_j + \beta)/(n + k\beta)$.

The most widely-studied class of distributions is the one that includes all of them: the $k$−dimensional simplex, $\Delta_k := \{(p_1, p_2, \ldots, p_k), : \sum_i p_i = 1, \ p_i \geq 0 \ \forall i \in [k]\}$. In the low-dimensional scaling, when $n/k \to \infty$ (the "dimension" here being the support size $k$), the minimax risk is

$$r_n(\Delta_k) = (1 + o(1))\, \frac{k - 1}{2n},$$

In [BS04], a variant of the add-constant estimator is shown to achieve this risk even to the constant. Furthermore, any add-constant estimator is rate optimal when $k$ is fixed. But in the very high-dimensional setting, when $k/n \to \infty$, [Pan04] showed that the minimax risk behaves as

$$r_n(\Delta_k) = (1 + o(1))\, \log \frac{k}{n},$$

achieved by an add-constant estimator, but with a constant that depends on the ratio of $k$ and $n$.

Despite these classical results on minimax optimal estimators, in practice people often use other estimators that have better empirical performance. This was a long-running mystery in the language modeling community [CG96], where variants of the Good–Turing estimator were shown to perform the best [JM85, GS95]. The gap in performance was only understood recently, using the notion of competitivity [OS15]. In essence, the Good–Turing estimator works well in *both* low- and

high-dimensional regimes, and in-between. Another estimator, *absolute discounting*, unlike add-constant estimators, simply *subtracts* a positive constant from the empirical counts and redistributes the subtracted amount to unseen categories. For a discount parameter $\delta \in [0, 1)$, it is defined as:

$$q_j^{-\delta} := \begin{cases} \frac{\mu_j - \delta}{n} & \text{if } \mu_j > 0, \\ \frac{D\delta}{n(k-D)} & \text{if } \mu_j = 0. \end{cases} \quad (1)$$

Starting with the work of [NEK94], absolute discounting soon supplanted the Good–Turing estimator, due to both its simplicity and comparable performance. Kneser-Ney smoothing [KN95], which uses absolute discounting at its core was long held as the preferred way to train $N$-gram models. Even to this day, the state-of-the-art language models are combined systems where one usually interpolates between recurrent neural networks and Kneser-Ney smoothing [JVS+16]. Can this success be explained?

Kneser-Ney is for the most part a principled implementation of the notion of back-off, which we only touch upon in the conclusion. The use of absolute discounting is critical however, as performance deteriorates if we back-off with care but use a more naïve add-constant or even Katz-style smoothing [Kat87], which switches from the Good–Turing to the empirical distribution at a fixed frequency point. It is also important to mention the Bayesian approach of [Teh06] that performs similarly to Kneser-Ney, called the Hierarchical Pitman-Yor language model. The hierarchies in this model reprise the role of back-off, while the two-parameter Poisson-Dirichlet prior proposed by Pitman and Yor [PY97] results in estimators that are very similar to absolute discounting. The latter is not a surprise because this prior almost surely generates a power law distribution, which is intimately related to absolute discounting as we study in this paper. Though our theory applies more generally, it can in fact be straightforwardly adapted to give guarantees to estimators built upon this prior.

## 4 Contributions

We investigate the reason behind the auspicious behavior of the absolute discounting estimator. We achieve this by demonstrating the adaptivity and competitivity of this estimator for many relevant families of distribution classes. In summary:

- We analyze the performance of the absolute discounting estimator by upper bounding the KL-risk for each class in a family of distribution classes defined by the expected number of distinct categories. [Section 5, Theorem 1] This result implies that absolute discounting achieves classical minimax rate-optimality in *both* the low- and high-dimensional regimes over the whole simplex $\Delta_k$, as outlined in Section 2.

- We provide a generic lower bound to the minimax risk of classes defined by a single distribution and all of its permutations. We then show that if the defining distribution is a truncated (possibly perturbed) power-law, then this lower bound matches the upper bound of absolute discounting, up to a constant factor. [Section 6, Corollaries 3 and 4]

- This implies that absolute discounting is adaptive to the family of classes defined by a truncated power-law distribution and its permutations. Also, since classes defined by the expected number of distinct categories necessarily includes a power-law, absolute discounting is also adaptive to this family. This is a strict refinement of classical minimax rate-optimality.

- We give an equivalence between the absolute discounting and Good–Turing estimators in the high-dimensional setting, whenever the distribution is a truncated power-law. This is a finite-sample guarantee, as compared to the asymptotic version of [OD12]. As a consequence, absolute-discounting becomes competitive with respect to the family of classes defined by permutations of power-laws, inheriting Good–Turing's behavior [OS15]. [Section 7, Lemma 5 and Theorem 6]

We corroborate the theoretical results with synthetic experiments that reproduce the theoretical minimax risk bounds. We also show that the prowess of absolute discounting on real data is not restricted only to language modeling. In particular, we explore a striking application to forecasting global terror incidents and show that, unlike naive estimators, absolute discounting gives accurate predictions simultaneously in all of low-, medium-, and high-activity zones. [Section 8]

## 5 Upper bound and classical minimax optimality

We now give an upper bound for the risk of the absolute discounting estimator and show that it recovers classical minimax rates in the low- and high-dimensional regimes. Recall that $d := \mathbb{E}[D]$ is the expected number of distinct categories in the samples. The upper bound that we derive can be written as function of only $d$, $k$, and $n$, and is non-decreasing in $d$. For a given $n$ and $k$, let $\mathcal{P}_d$ be the set of all distributions for which $\mathbb{E}[D] \leq d$. The upper bound is thus also a worst-case bound over $\mathcal{P}_d$.

**Theorem 1** (Upper bound). *Consider the absolute discounting estimator $q = q^{-\delta}$, defined in* (1). *Let $p$ be such that $\mathbb{E}[D] = d$. Given a discount $0 < \delta < 1$, there exists a constant $c$ that may depend on $\delta$ and only $\delta$, such that*

$$r_n(p,q) \leq \begin{cases} \dfrac{d}{n} \log \dfrac{k - \frac{d}{2}}{\frac{d}{2}} + c\dfrac{d}{n} & \text{if} \quad d \geq 10 \log \log k, \\ \dfrac{d}{n} \log k + c\dfrac{d}{n} & \text{if} \quad d < 10 \log \log k. \end{cases} \tag{2}$$

*The same bound holds for $r_n(\mathcal{P}_d, q)$.*

We defer the proof the theorem to the supplementary material. Here are the immediate implications. For the low-dimensional regime $\frac{n}{k} \to \infty$ and the class $\Delta_k$, the largest $d$ can be once $n > k$ is $k$. The risk of absolute discounting is thus bounded by $c(1 + o(1))\frac{k}{n} = \mathcal{O}(1)\frac{k}{n}$. This is minimax rate-optimal [BS04]. For the high-dimensional regime $\frac{k}{n} \to \infty$ and the class $\Delta_k$, the largest $d$ can be when $k > n$ is $n$. The risk of absolute discounting is thus dominated by the first term, which reduces to $(1 + o(1)) \log \frac{k}{n}$. This is the optimal risk for the class $\Delta_k$ [Pan04], even to the constant.

Therefore on the two extreme ranges of $k$ and $n$ absolute discounting recovers the best performance, either as rate-optimal or optimal even to the constant. These results are for the entire $k-$dimensional simplex $\Delta_k$. Furthermore, for smaller classes, it characterizes the worst-case risk of the class by the $d$, the expected number of distinct categories. Is this characterization tight?

## 6 Lower bounds and adaptivity

In order to lower bound the minimax risk of a given class $\mathcal{P}$, we use a finer granularity than the $\mathcal{P}_d$ classes described in Section 5. In particular, let $\mathcal{P}_p$ be the *permutation class* of distributions consisting of a single distribution $p$ and all of its permutations. Note that the multiset of probabilities is the same for all distributions in $\mathcal{P}_p$, and since the expected number of distinct categories only depends on the multiset ($d = \sum_j [1 - (1 - p_j)^n]$) it follows that $\mathcal{P}_p \subset \mathcal{P}_d$[1]. To find a good lower bound for $\mathcal{P}_d$, we need a $p$ that is "worst case". We first give the following generic lower bound.

**Theorem 2** (Generic lower bound). *Let $\mathcal{P}_p$ be a permutation class defined by a distribution $p$ and let $\gamma > 1$. Then for $k > \gamma d$, the minimax risk is bounded by:*

$$r_n(\mathcal{P}_p) \geq \left(1 - \frac{1}{\gamma}\right)\left(\sum_{j=\gamma d}^{k} p_j\right) \log \frac{k - \gamma d}{\sum_{j=\gamma d}^{k} p_j} + \sum_{i=\gamma d} p_j \log p_j \tag{3}$$

Equation (3) can be used as a starting point for more concrete lower bounds on various distribution classes. We illustrate this for two cases. First, let us choose $p$ to be a truncated power-law distribution with power $\alpha$: $p_j \propto j^{-\alpha}$, for $j = 1, \cdots, k$. We always assume $\alpha \geq \alpha_0 > 1$. This leads to the following lower bound.

**Corollary 3.** *Let $\mathcal{P}$ be all permutations of a single power-law distribution with power $\alpha$ truncated over $k$ categories. Then there exists a constant $c > 0$ and large enough $n_0$ such that when $n > n_0$ and $k > \max\{n, 1.2^{\frac{1}{\alpha-1}} n^{\frac{1}{\alpha}}\}$,*

$$r_n(\mathcal{P}) \geq c\frac{d}{n} \log \frac{k - 2d}{2d}.$$

Next, we use a different choice of $p$ for $\mathcal{P}_p$ to provide a lower bound whenever $d$ grows linearly with $n$. This essentially closes the gap of the previous corollary when $\alpha$ approaches 1.

**Corollary 4.** *Let $\rho \in (1, 1.75)$ and let $\mathcal{P}$ be all permutations of a single uniform distribution over a subset $k' = \frac{n}{\rho}$ out of $k$ categories. Then $d \sim (1 - e^{-\rho})n/\rho$ and there exists a constant $c > 0$ and large enough $n_0$ such that when $n > n_0$ and $k > n^5$,*

$$r_n(\mathcal{P}) \geq c\frac{d}{n}\log\frac{k - 1.2d}{d} \ .$$

We defer the proofs of the theorem and its corollaries to the supplementary material. The upper bound of Theorem 1 and the lower bounds of Corollaries 3 and 4 are within constant factors of each other. The immediate consequence is that absolute discounting is adaptive with respect to the families of classes of the Corollaries. Furthermore, over the family of classes $\mathcal{P}_d$ where we can write $d$ as $n^{\frac{1}{\alpha}}$ for some $\alpha > 1$ or $d \propto n$, we can select a distribution from the Corollaries among each class and use the corresponding lower bound to match the upper bound of Theorem 1 up to a constant factor. Therefore absolute discounting is adaptive to this family of classes. Intuitively, adaptivity to these classes establishes optimality in the intermediate range between low- and high-dimensional settings in a distribution-dependent fashion and governed by the expected number of distinct categories $d$, which we may regard as the *effective* dimension of the problem.

## 7   Relationship to Good–Turing and competitivity

We now establish a relationship between the absolute discounting and Good–Turing estimators and refine the adaptivity results of the previous section into competitivity results. When [OS15] introduced the notion of competitive optimality, they showed that a variation of the Good–Turing estimator is worst-case competitive with respect to the family of distribution classes defined by any given probability distribution and its permutations. In light of the results of Sections 5 and 6, it is natural to ask whether absolute discounting enjoys the same kind of competitive properties. Not only that, but it was observed empirically by [NEK94] and shown theoretically in [OD12] that *asymptotically* Good–Turing behaves exactly like absolute discounting, when the underlying distribution is a (possibly perturbed) power-law. We therefore choose this family of classes to prove competitivity for. We first make the aforementioned equivalence concrete by establishing a *finite sample* version. We use the following *idealized* version of the Good–Turing estimator [Goo53]:

$$q_j^{\mathsf{GT}} := \begin{cases} \frac{\mu_j + 1}{n}\frac{\mathbb{E}[\Phi_{\mu_j + 1}]}{\mathbb{E}[\Phi_{\mu_j}]} & \text{if } \mu_j > 0, \\ \frac{\mathbb{E}[\Phi_1]}{n(k - D)} & \text{if } \mu_j = 0. \end{cases} \tag{4}$$

**Lemma 5.** *Let $p$ be a power law with power $\alpha$ truncated over $k$ categories. Then for $k > \max\{n, n^{\frac{1}{\alpha - 1}}\}$, we have the equivalence:*

$$q_j^{\mathsf{GT}} = \frac{\mu_j - \frac{1}{\alpha}}{n}\left(1 + \mathcal{O}\left(n^{-\frac{1}{2}\frac{3}{2\alpha + 1}}\right)\right) \sim \frac{\mu_j - \frac{1}{\alpha}}{n} \qquad \forall \mu_j \in \left\{1, \cdots, n^{\frac{1}{2\alpha + 1}}\right\} \ .$$

An interesting outcome of the equivalence of Lemma 5 is that it suggests a choice of the discount $\delta$ in terms of the power, $1/\alpha$. To give a data-driven version of $1/\alpha$, we will use a robust version of the ratio $\Phi_1/D$ proposed in [OD12, BBO17], which is a strongly consistent estimator when $k = \infty$.

**Theorem 6.** *Let $\mathcal{P}$ be all permutations of a truncated power law $p$ with power $\alpha$. Let $q$ be the absolute discounting estimator with $\delta = \min\left\{\frac{\max\{\Phi_1, 1\}}{D}, \delta_{\max}\right\}$, for a suitable choice of $\delta_{\max}$. Then for $k > \max\{n, n^{\frac{1}{\alpha - 1}}\}$, the competitive loss is*

$$\epsilon_n(\mathcal{P}_p, q) = \mathcal{O}\left(n^{-\frac{2\alpha - 1}{2\alpha + 1}}\right) \ .$$

The implications are as follows. For the union of all such classes above a given $\alpha$, we find that we beat the $n^{-1/3}$ rate of the worst-case competitive loss obtained for the estimator in [OS15]. Theorem 6 and the bounds of Sections 5 and 6, together imply that absolute discounting is not only worst-case competitive, but also *class-by-class* competitive with respect to the power-law permutation family. In other words, it in fact achieves minimax optimality even to the constant. One of the advantages of absolute discounting is that it gradually transitions between values that are close to the empirical distribution for abundant categories (since $\mu$ then dominates the discount $\delta$), to a behavior that imitates the Good–Turing estimator for rare categories (as established by Lemma 5). In contrast, the estimator proposed in [OS15], and its antecedents starting from [Kat87], have to carefully choose a threshold where they switch abruptly from one estimator to the other.

# 8 Experiments

We now illustrate the theory with some experimental results. Our purpose is to (1) validate the functional form of the risk as given by our lower and upper bounds and (2) compare absolute discounting on both synthetic and real data to estimators that have various optimality guarantees. In all synthetic experiments, we use 500 Monte Carlo iterations. Also, we set the discount value based on data, $\delta = \min\{\frac{\max(\Phi_1, 1)}{D}, 0.9\}$. This is as suggested in Section 7, assuming $\delta_{\max} = 0.9$ is sufficient.

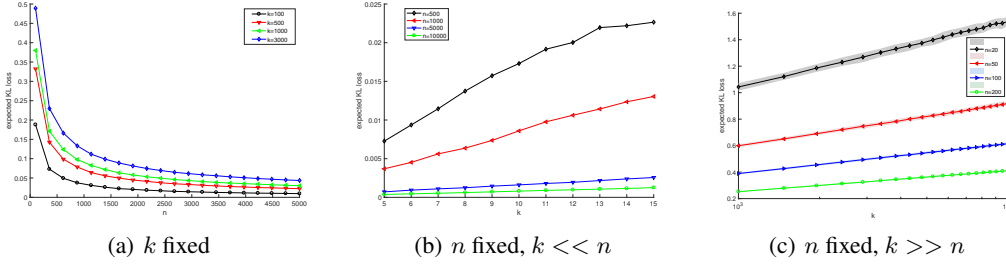

|(a) $k$ fixed|(b) $n$ fixed, $k << n$|(c) $n$ fixed, $k >> n$|

Figure 1: Risk of absolute discounting in different ranges of $k$ and $n$ for a power-law with $\alpha = 2$

**Validation**  For our first goal, we consider absolute discounting in isolation. Figure 1(a) shows the decay of KL-risk with the number of samples $n$ for a power-law distribution. The dependence of the risk on the number of categories $k$ is captured in Figures 1(b) (linear $x$-axis) and 1(c) (logarithmic $x$-axis). Note the linear growth when $k$ is small and the logarithmic growth when $k$ is large. For the last plot we give 95% confidence intervals for the simulations, by performing 100 restarts.

**Synthetic data**  For our second goal, we start with synthetic data. In Figure 2, we pit absolute discounting against a number of distributions related to power-laws. The estimators used for our comparisons are: empirical $q^{+0}(x) = \frac{\mu_x}{n}$, add-beta $q^{+\beta}(x) = \frac{\mu_x + \beta_{\mu_x}}{N}$, and its two variants:

- Braess and Sauer, $q^{\mathsf{BS}}$ [BS04] $q^{+\beta}$ with $\beta_0 = 0.5$, $\beta_1 = 1$, and $\beta_i = 0.75 \ \forall i \geq 2$
- Paninski, $q^{\mathsf{Pan}}$ [Pan04] $q^{+\beta}$ with $\beta_i = \frac{n}{k} \log \frac{k}{n} \ \forall i$,

absolute discounting, $q^{-\delta}$, described in 1, Good–Turing + empirical $q^{\mathsf{GT}}$ in [OS15], and an oracle-aided estimator where $S_\mu$ is known.

In Figures 2(a) and 2(b), samples are generated according to a power-law distribution with power $\alpha = 2$ over $k = 1,000$ categories. However, the underlying distribution in Figure 2(c) is a piecewise power-law. It consists of three equal-length pieces, with powers 1.3, 2, and 1.5. Paninski's estimator is not shown in Figures 2(b) and 2(c) since it is not well-defined in this range (it is designed for the case $k > n$ only). Unsurprisingly, absolute discounting dominates these experiments. What is more interesting is that it does not seem to need a pure power-law (similar results hold for other kinds of perturbations, such as mixtures and noise). Also Good–Turing is a tight second.

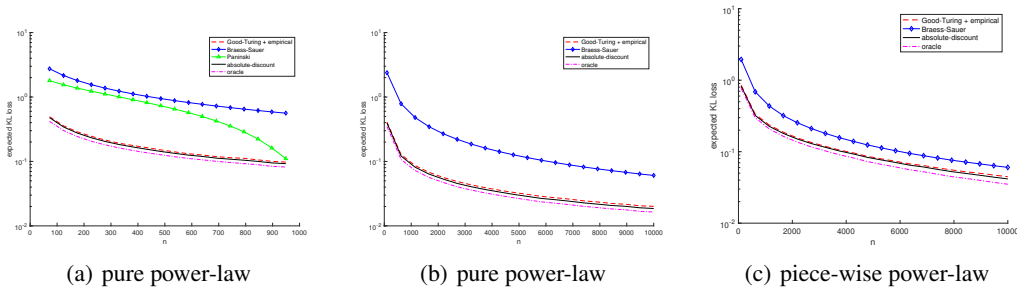

|(a) pure power-law|(b) pure power-law|(c) piece-wise power-law|

Figure 2: Comparing estimators for power-law variants with power $\alpha = 2$ and $k = 1000$.

**Real data** One of the chief motivations to investigate absolute discounting is natural language modeling. But there have been such extensive empirical studies that have verified over and over the power of absolute discounting (see the classical survey of [CG96]) that we chose to use this space for something new. We use the START *Global terrorism database* from the University of Maryland [LDMN16] and explore how well we can forecast the number of terrorist incidents in different cities. The data contains the record of more than $50,000$ terror incidents between the years 1992 and 2010, in more than $12,000$ different cities around the world. First, we display in Figure 3(a) the frequency of incidents across the entire dataset versus the activity rank of the city in log-log scale, showing a striking adherence to a power-law (see [CSN09] for more on this).

The forecasting problem that we solve is to estimate the number of total incidents in a subset of the cities over the coming year, using the current year's data from all cities. In order to emulate the various dimension regimes, we look at three subsets: (1) low-activity cities with *no* incidents in the current year and less than 20 incidents in the whole data, (2) medium-activity cities, with *some* incidents in the current year and less than 20 incidents in the whole data, and (3) high-activity individual cities with a large number of overall incidents.

The results for (1) are in Figure 3(b). The frequency estimator trivially estimates zero. Braess-Sauer does something meaningful. But absolute discounting and Good–Turing estimators, indistinguishable from each other, are remarkably on spot. And this, without having observed any of the cities! This nicely captures the importance of using structure when dimensionality is so high and data is so scarce. The results for (2) are in Figure 3(c). The frequency estimator markedly overestimates. But now absolute discounting, Good–Turing, and Braess-Sauer, perform similarly. This is a lower dimensional regime than in (1), but still not adequate for simply using frequencies. This changes in case (3), illustrated in Figure 4. To take advantage of the abundance of data, in this case at each time point we used the previous $2,000$ incidents for learning, and predicted the share of each city for the next $2,000$ incidents. In fact, incidents are so abundant that we can simply rely on the previous window's count. Note how Braess-Sauer over-penalizes such abundant categories and suffers, whereas absolute discounting and Good–Turing continue to hold their own, mimicking the performance of the empirical counts. This is a very low-dimensional regime.

The closeness of the Good–Turing estimator to absolute discounting in all of our experiments validates the equivalence result of Lemma 5. The robustness in various regimes and the improvement in performance over such minimax optimal estimators as Braess-Sauer's and Paninski's are evidence that absolute discounting truly molds to both the raw dimension and effective dimension / structure.

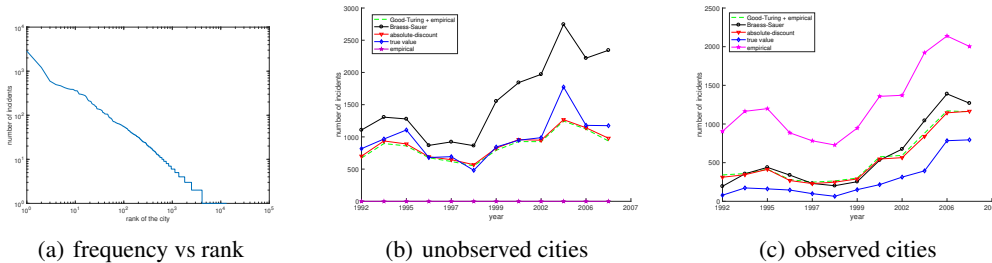

|              |              |              |
| :----------: | :----------: | :----------: |
| (a) frequency vs rank | (b) unobserved cities | (c) observed cities |

Figure 3: $(a)$ power-law behavior of frequency vs rank in terror incidents, $(b)$, and $(c)$ comparing forecasts of the number of incidents in unobserved cities and observed ones, respectively.

## 9    Conclusion

In this paper, we offered a rigorous analysis of the absolute discounting estimator for categorical distributions. We showed that it recovers classical minimax optimality. The true reason for its success, however, is in adapting to distributions much more intimately, by recovering the right dependence on the distinct observed categories $d$, which can be regarded as an effective dimension, and optimally tracking structure such as power-laws. We also tightened its relationship with the celebrated Good–Turing estimator.

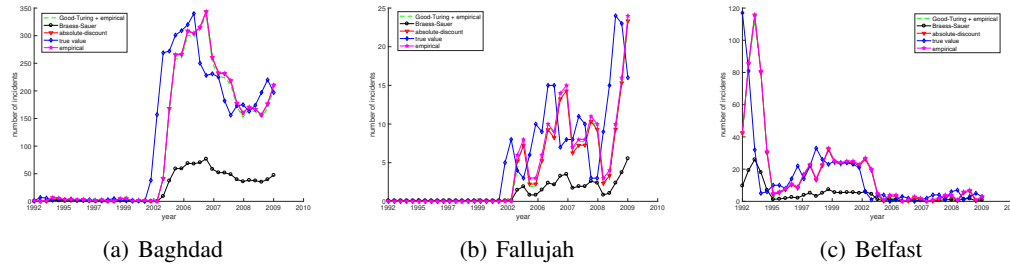

|  (a) Baghdad | (b) Fallujah | (c) Belfast |

Figure 4: Estimating the number of incidents based on previous data for different cities

Some of our analysis could possibly be tightened, in particular in terms of the range of applicability over $n$, $k$, and $d$. Also, the limiting case of $\alpha = 1$ (very heavy tails, known as "fast variation" [BBO17]) to which our results don't directly apply, merits investigation. But more importantly, absolute discounting is often a module. For example, we already note how it is widely used in $N$-gram back-off models [KN95]. Also, recently, it has been successfully applied to smoothing low-rank probability matrices [FOO16]. Perhaps to further understand its power, it is worthwhile to study how it interacts with such larger systems.

**Acknowledgements**    We thank Vaishakh Ravindrakumar for very helpful suggestions, and NSF for supporting this work through grants CIF-1564355 and CIF-1619448.

## Footnotes

[1]We abuse notation by distinguishing the classes by the letter used, while at the same time using the letters to denote actual quantities. From the context we understand that $d$ is the expected number of distinct categories for $p$, at the given $n$.

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
