[Supplementary Material]

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

# A  Proof of Theorem 1, upper bound

We start with a technical note. Though we presented the framework for a fixed sample size $n$, the entirety of the paper analyzes the "Poissonized" version. In the Poisson sampling model, the number of samples is in fact $N \sim \text{POI}(n)$, a Poisson random variable with mean $n$. This is often the more natural model when data is collected within a fixed time window, in contrast to until a certain number of samples are collected. Or we can think of Poisson sampling as a convenience because it makes all counts independent and distributed according to $\mu_j \sim \text{POI}(np_j)$. It is possible to "de-Poissonize" the results, but we omit this for brevity.

In proof of the theorem, we show a more general upper bound. We upper bound the instantaneous risk of a class of distributions based on $d$, $\mathbb{E}[\Phi_1]$, and $\mathbb{E}[\Phi_2]$, the expected number of distinct categories, categories that appeared once, and twice respectively. Namely, we show for some constant $c$,

$$\max_{p \in \mathcal{P}_d} \mathbb{E}_{x^n} \left[ \mathsf{KL}(p||q(x^n)) \right] \leq \frac{\mathbb{E}[\Phi_1]}{n} \log \frac{2k - d}{d\delta} + \frac{\mathbb{E}[\Phi_1]}{n} + \frac{2\mathbb{E}[\Phi_2]}{n} \log \frac{1}{1 - \delta} + \frac{c \cdot d}{n}.$$

*Proof.*

$$\mathbb{E}_{X^n \sim p^n} \left[ \mathsf{KL}(p||q(X^n)) \right]$$

$$= \mathbb{E}_{X^n \sim p^n} \left[ \sum_{j=1}^{k} p_j \log \frac{p_j}{q_j(X^n)} \right]$$

$$= \mathbb{E} \left[ \sum_{j=1}^{k} \mathbb{1}_j^0 p_j \log \frac{np_j(k-D)}{D\delta} + \sum_{j=1}^{k} \sum_{i=1}^{\infty} \mathbb{1}_j^i p_j \log \frac{np_j}{i - \delta} \right]$$

$$= \mathbb{E} \left[ \sum_{j=1}^{k} \mathbb{1}_j^0 p_j \log np_j + \mathbb{1}_j^0 p_j \log \frac{(k-D)}{D\delta} + \sum_{j=1}^{k} \sum_{i=1}^{\infty} \mathbb{1}_j^i p_j \log \frac{np_j}{i - \delta} \right]$$

$$\overset{(a)}{=} \frac{1}{n} \sum_{j=1}^{k} e^{-\lambda_j} \lambda_j \log \lambda_j + \frac{1}{n} \sum_{j=1}^{k} \lambda_j \mathbb{E} \left[ \mathbb{1}_j^0 \log \frac{k-D}{D\delta} \right] + \frac{1}{n} \sum_{j=1}^{k} \sum_{i=1}^{\infty} \lambda_j \log \frac{\lambda_j}{i - \delta} \text{poi}(np_j, i)$$

$$\overset{(b)}{=} \frac{1}{n} \sum_{j=1}^{k} \lambda_j \mathbb{E} \left[ \mathbb{1}_j^0 \log \frac{k-D}{D\delta} \right] + \frac{1}{n} \sum_{j=1}^{k} \left( \lambda_j \log \lambda_j + \sum_{i=1}^{\infty} \lambda_j \log \frac{1}{i - \delta} \text{poi}(\lambda_j, i) \right) \qquad (5)$$

where $(a)$ is by Poisson sampling and replacing $\lambda_j := np_j$, and $(b)$ is by combining the first and last expressions. Now we state two lemmas that are helpful in bounding each of the two terms in (5).

**Lemma 7.** *For all $p \in \mathcal{P}_d$ and with the assumption of $D > 2$, for $d > 10 \log \log k$,*

$$\mathbb{E}_{X^n \sim p^n} \left[ \mathbb{1}_j^0 \log \frac{k-D}{D} \right] \leq e^{-\lambda_j} \left( 1 + \log \frac{k - \frac{d}{2}}{\frac{d}{2}} \right)$$

*and for $d < 10 \log \log k$,*

$$\mathbb{E}_{X^n \sim p^n} \left[ \mathbb{1}_j^0 \log \frac{k-D}{D} \right] \leq e^{-\lambda_j} \log k.$$

**Lemma 8.** *For $x > 0$ and for $0 < \delta < 1$, $x \log x + \sum_{i=2}^{\infty} x \log \frac{1}{i-\delta} \text{poi}(x, i) < c'$ for some constant $c'$.*

We can write the second part in (5) as

$$\frac{1}{n} \sum_{j=1}^{k} \lambda_j \log \frac{1}{1 - \delta} \text{poi}(\lambda_j, 1) + \frac{1}{n} \sum_{j=1}^{k} \left( \lambda_j \log \lambda_j + \sum_{i=2}^{\infty} \lambda_j \log \frac{1}{i - \delta} \text{poi}(\lambda_j, i) \right), \qquad (6)$$

and since the second term in (6) is negative for all $\lambda_j < 1$, (6) is upper bounded by

$$\frac{1}{n}\sum_{j=1}^{k}\lambda_j^2 e^{-\lambda_j}\log\frac{1}{1-\delta} + \frac{1}{n}\sum_{\lambda_j\geq 1}\left(\lambda_j\log\lambda_j + \sum_{i=2}^{\infty}\lambda_j\log\frac{1}{i-\delta}\text{poi}(\lambda_j,i)\right).$$

Continuing from (5), using Lemmas 7, 8 and the definitions of $\mathbb{E}[\Phi_1] = \sum_{j=1}^{k}e^{-\lambda_j}\lambda_j$ and $\mathbb{E}[\Phi_2] = \sum_{j=1}^{k}e^{-\lambda_j}\frac{\lambda_j^2}{2}$,

$$\mathop{\mathbb{E}}_{X^n\sim p^n}[\text{KL}(p\|q(X^n))] \leq \frac{\mathbb{E}[\Phi_1]}{n}\left(\log\frac{2k-d}{d\delta}+1\right) + \frac{2\mathbb{E}[\Phi_2]}{n}\log\frac{1}{1-\delta} + \frac{1}{n}\sum_{j:\lambda_j\geq 1}c'$$

$$\leq \frac{\mathbb{E}[\Phi_1]}{n}\left(\log\frac{2k-d}{d\delta}+1\right) + \frac{2\mathbb{E}[\Phi_2]}{n}\log\frac{1}{1-\delta} + \frac{c'\cdot d}{n(1-e^{-1})},$$

where the last line is because $d = \sum_j(1-e^{\lambda_j}) \geq \sum_{\lambda_j\geq 1}(1-e^{\lambda_j}) \geq |\{j:\lambda_j\geq 1\}|(1-e^{-0.7})$ $\square$

## A.1 Proof of Lemma 7

*Proof.* Using Lemma 17,

$$\mathop{\mathbb{E}}_{X^n\sim p^n}[\mathbb{1}_j^0\log\frac{k-D}{D}] = e^{-\lambda_j}\mathbb{E}\left[\log\frac{k-D}{D}\Big| D < d-\sqrt{2vs},\mu_j=0\right]\Pr(D < d-\sqrt{2vs})$$

$$+ e^{-\lambda_j}\mathbb{E}\left[\log\frac{k-D}{D}\Big| D > d-\sqrt{2vs},\mu_j=0\right]\Pr(D > d-\sqrt{2vs})$$

$$\leq e^{-\lambda_j}\left(e^{-s}\log(k-1) + \log\frac{k-d+\sqrt{2vs}}{d-\sqrt{2vs}}\right).$$

Choosing $s = \log\log k$ and assuming $D \geq 2$ and $d > 10\log\log k$ yield the results. Note that if $\mu_j = 0$, it can change $D$ by at most one and its effect can be ignored. Also when $d < 10\log\log k$ we can use the naive bound of $\log k$, since $\log\frac{k-D}{D} < \log k$ for $D > 1$. $\square$

## A.2 Proof of Lemma 8

*Proof.* We first assume $x > 100$ and prove the lemma.

$$\sum_{i=2}^{\infty}\text{poi}(x,i)\log(i-\delta) \tag{7}$$

$$\geq \sum_{i=x-x_0}^{x+x_0}\text{poi}(x,i)\log(i-\delta)$$

$$= \text{poi}(x,x)\log(x-\delta) + \sum_{a=1}^{x_0}\text{poi}(x,x-a)\log(x-a-\delta) + \text{poi}(x,x+a)\log(x+a-\delta)$$

$$\geq \text{poi}(x,x)\log(x-\delta) + \sum_{a=1}^{x_0}\text{poi}(x,x-a)\Big[\log(x-a-\delta)+\log(x+a-\delta)\Big]$$

$$= \sum_{a=0}^{x_0}\frac{\text{poi}(x,x-a)+\text{poi}(x,x+a)}{2}\Big[\log(x-a-\delta)+\log(x+a-\delta)\Big]$$

$$+ \sum_{a=1}^{x_0}\frac{\text{poi}(x,x-a)-\text{poi}(x,x+a)}{2}\Big[\log(x-a-\delta)+\log(x+a-\delta)\Big] \tag{8}$$

By Lemma 18,

$$\sum_{a=0}^{x_0}\text{poi}(x,x-a)+\text{poi}(x,x+a) = \text{poi}(x,x)+1-\Pr(\text{POI}(x)>x+x_0)-\Pr(\text{POI}(x)<x-x_0)$$

$$\geq \frac{1}{e\sqrt{x}}+1-2\cdot e^{x_0-(x+x_0)\ln(1+\frac{x_0}{x})},$$

Also we can lower bound the bracket in (8) as

$$
\begin{aligned}
\log(x + a - \delta) + \log(x - a + \delta) &= \log\left((x-\delta)^2 - a^2\right) \\
&= \log(x^2 - 2x\delta + \delta^2 - a^2) \\
&= \log\left(x^2(1 - \frac{2\delta}{x} + \frac{\delta^2 - a^2}{x^2})\right) \\
&= 2\log x + \log(1 - \frac{2\delta}{x} + \frac{\delta^2 - a^2}{x^2}) \\
&\geq 2\log x - \frac{4\delta}{x} - \frac{2(a^2 - \delta^2)}{x^2}.
\end{aligned}
$$

Thus for some constant $c_1$ and $x_0 = x^{0.8}$,

$$
\begin{aligned}
\sum_{a=0}^{x_0} & \frac{\text{poi}(x, x-a) + \text{poi}(x, x+a)}{2}\left[\log(x - a - \delta) + \log(x + a - \delta)\right] \\
&\geq \left(1 + \frac{1}{e\sqrt{x}} - 2e^{x_0 - (x+x_0)\ln(1 + \frac{x_0}{x})}\right)\left(\log x - \frac{2\delta}{x}\right) \\
&\quad - \sum_{a=0}^{x_0}\left(\text{poi}(x, x-a) + \text{poi}(x, x+a)\right)\left(\frac{a^2}{x^2}\right) \\
&= \log x - \frac{2\delta}{x} - 2e^{x_0 - (x+x_0)\ln(1 + \frac{x_0}{x})}(\log x - \frac{2\delta}{x}) \\
&\quad - \sum_{a=0}^{x_0}\left(\text{poi}(x, x-a) + \text{poi}(x, x+a)\right)\left(\frac{a^2}{x^2}\right) \\
&\geq \log x - \frac{c_1}{x}.
\end{aligned}
\tag{9}
$$

where the last line is due to the following lemma.

**Lemma 9.** *For $x_0 = x^{0.8}$ there exists a constant $c_1$ such that*

$$
\sum_{a=0}^{x_0}\left[\text{poi}(x, x-a) + \text{poi}(x, x+a)\right](\frac{a^2}{x^2}) \leq \frac{c_1}{x}.
$$

The difference in probabilities of two equidistant points from the mean of a Poisson distribution is bounded by

$$
\begin{aligned}
\text{poi}(x, x+a) - \text{poi}(x, x-a) &= \frac{e^{-x}x^{x-a}}{(x-a)!}\left[\frac{1}{(1 + \frac{a}{x})(1 + \frac{a-1}{x})\ldots(1 + \frac{1-a}{x})} - 1\right] \\
&= \frac{e^{-x}x^{x-a}e^{x-a}}{(x-a)^{x-a}\sqrt{2\pi(x-a)}}\left[\frac{1}{(1 + \frac{a}{x})(1 + \frac{a-1}{x})\ldots(1 + \frac{1-a}{x})} - 1\right] \\
&= \frac{e^{-a}}{\sqrt{2\pi(x-a)}}\left[\frac{1}{(1 + \frac{a}{x})(1 + \frac{a-1}{x})\ldots(1 + \frac{1-a}{x})} - 1\right] \\
&\approx \frac{e^{-a}}{\sqrt{2\pi(x-a)}}\frac{4}{x},
\end{aligned}
$$

and therefore for $x_0 = x^{0.8}$ and some constant $c_5$,

$$\sum_{a=1}^{x_0} \frac{\text{poi}(x, x-a) - \text{poi}(x, x+a)}{2} \Big[ \log(x - a - \delta) + \log(x + a - \delta) \Big]$$

$$\geq -\sum_{a=1}^{x_0} \frac{e^{-a}}{\sqrt{2\pi(x-a)}} \frac{2}{x} \log\left((x-\delta)^2 - a^2\right)$$

$$\geq -\sum_{a=1}^{x_0} \frac{e^{-a}}{\sqrt{2\pi(x-a)}} \frac{2}{x} \log x^2$$

$$\geq -\frac{\sum_{a=1}^{\infty} e^{-a}}{\sqrt{2\pi(x-x_0)}} \frac{4}{x} \log x$$

$$\geq -\frac{4 \log x}{x\sqrt{2\pi(x-x_0)}}$$

$$\geq -\frac{c_5}{x}. \tag{10}$$

Selecting $c > c_1 + c_5$ leads to the Lemma. It can be shown that the lemma is valid for $x < 100$ by plotting the function. $\qquad\square$

### A.3 Proof of Lemma 9

*Proof.*

$$\sum_{a=0}^{x_0} \Big[ \text{poi}(x, x-a) + \text{poi}(x, x+a) \Big] (\frac{a^2}{x^2})$$

$$\leq \sum_{a=0}^{x_0} \frac{2a^2}{x^2} \text{poi}(x, x-a)$$

$$= \sum_{a=0}^{x_0} \frac{2a^2}{x^2} \frac{e^{-x} x^{x-a}}{(x-a)!}$$

$$\overset{(a)}{\leq} \sum_{a=0}^{x_0} \frac{2a^2}{x^2} \left[ \frac{e^{-x+x-a} x^{x-a}}{(x-a)^{x-a} \sqrt{2\pi(x-a)}} \right]$$

$$= \sum_{a=0}^{x_0} \frac{2a^2}{x^2} \left[ \frac{e^{-a}}{\sqrt{2\pi(x-a)}} \left( 1 + \frac{a}{x-a} \right)^{x-a} \right]$$

$$= \sum_{a=0}^{x_0} \frac{2a^2}{x^2} \left[ \frac{e^{-a}}{\sqrt{2\pi(x-a)}} e^{(x-a)\ln(1+\frac{a}{x-a})} \right]$$

$$\overset{(b)}{\leq} \sum_{a=0}^{x_0} \frac{2a^2}{x^2} \left[ \frac{e^{-\frac{a^2}{4(x-a)}}}{\sqrt{2\pi(x-a)}} \right],$$

where $(a)$ is by Stirling's approximation and $(b)$ is because $\ln(1+x) < x - \frac{x^2}{4}$ for $x < 1$. We can decompose the last summation to three different summations as

$$\sum_{a=0}^{x_0} \frac{2a^2}{x^2} \left[ \frac{e^{-\frac{a^2}{4(x-a)}}}{\sqrt{2\pi(x-a)}} \right]$$

$$= \sum_{a=0}^{\sqrt{x}} \frac{2a^2}{x^2} \left[ \frac{e^{-\frac{a^2}{4(x-a)}}}{\sqrt{2\pi(x-a)}} \right] + \sum_{a=\sqrt{x}+1}^{\sqrt{x}\ln x} \frac{2a^2}{x^2} \left[ \frac{e^{-\frac{a^2}{4(x-a)}}}{\sqrt{2\pi(x-a)}} \right] + \sum_{a=\sqrt{x}\ln x}^{x_0} \frac{2a^2}{x^2} \left[ \frac{e^{-\frac{a^2}{4(x-a)}}}{\sqrt{2\pi(x-a)}} \right] \tag{11}$$

Now we bound each term in (11). For the first term and for some constant $c_2$:

$$\sum_{a=0}^{\sqrt{x}} \frac{2a^2}{x^2}\left[\frac{e^{-\frac{a^2}{4(x-a)}}}{\sqrt{2\pi(x-a)}}\right] \leq 2\sqrt{x}\frac{x}{x^2}\frac{1}{\sqrt{2\pi(x-\sqrt{x})}}$$

$$\leq \sqrt{\frac{2}{\pi}}\frac{1}{x}\frac{1}{\sqrt{1-\frac{\sqrt{x}}{x}}}$$

$$\leq \sqrt{\frac{2}{\pi}}\frac{1}{x}(1+\frac{\sqrt{x}}{2x}) \leq \frac{c_2}{x}.$$

Also for the middle term in (11) and some constant $c_4$:

$$\sum_{a=\sqrt{x}+1}^{\sqrt{x}\ln x} \frac{2a^2}{x^2}\left[\frac{e^{-\frac{a^2}{4(x-a)}}}{\sqrt{2\pi(x-a)}}\right]$$

$$\leq \sqrt{\frac{2}{\pi}}\frac{1}{x^2}\frac{1}{\sqrt{x-\sqrt{x}\ln x}}\sum_{a=\sqrt{x}+1}^{\sqrt{x}\ln x}a^2e^{-\frac{a^2}{4x}}$$

$$\leq \sqrt{\frac{2}{\pi}}\frac{1}{x^2}\frac{1}{\sqrt{x-\sqrt{x}\ln x}}\int_{\sqrt{x}}^{\sqrt{x}\ln x}a^2e^{-\frac{a^2}{4x}}da$$

$$= \sqrt{\frac{2}{\pi}}\frac{1}{x^2}\frac{1}{\sqrt{x-\sqrt{x}\ln x}}2\left[x\sqrt{x}e^{-\frac{1}{4}}-x\sqrt{x}e^{-\frac{x\ln^2 x}{4x}}\ln x+2\sqrt{\pi}\left(\text{Erf}(\frac{\ln x}{2})-\text{Erf}(\frac{1}{2})\right)\right]$$

$$\leq \sqrt{\frac{2}{\pi}}\frac{1}{x^2\sqrt{x}}\frac{1}{\sqrt{1-\frac{\sqrt{x}\ln x}{x}}}(4x^{\frac{3}{2}}e^{-\frac{1}{4}})$$

$$\leq \frac{c_4}{x}.$$

Similarly for the third term in (11) and for some constant $c_3$, we can write

$$\sum_{a=\sqrt{x}\ln x}^{x_0} \frac{2a^2}{x^2}\left[\frac{e^{-\frac{a^2}{4(x-a)}}}{\sqrt{2\pi(x-a)}}\right] \leq (x_0-\sqrt{x}\ln x)\frac{2x_0^2}{x^2}\left[\frac{e^{-\frac{(\sqrt{x}\ln x)^2}{4x}}}{\sqrt{2\pi(x-x_0)}}\right]$$

$$\leq \sqrt{\frac{2}{\pi}}\frac{1}{x^2}\frac{x_0^3}{\sqrt{x-x_0}}e^{-\frac{\ln^2 x}{4}}$$

$$= \sqrt{\frac{2}{\pi}}\frac{1}{x^2}\frac{x_0^3}{\sqrt{x-x_0}}\frac{1}{x^{\frac{\ln x}{4}}} \leq \frac{c_3}{x}.$$

Choosing $c_1 \geq c_2+c_3+c_4$ leads to the lemma. $\qquad\qquad\square$

# B   Proofs of lower bound

In this part we provide the proofs of Theorem 2 as well as Corollaries 3 and 4. In order to lower bound the minimax risk of a given class $\mathcal{P}$, we can resort to two simplifications. First, we consider classes at a much a finer granularity than the $\mathcal{P}_d$ classes described in Section 5. In particular, let $\mathcal{P}_p$ be the *permutation class* of distributions consisting of a single distribution $p$ and all of its permutations. Note that the multiset of probabilities is the same for all distributions in $\mathcal{P}_p$, and since the expected number of distinct categories only depends on the multiset ($d = \sum_j[1-(1-p_j)^n]$) it follows that $\mathcal{P}_p \subset \mathcal{P}_d$. [2]. To find a good lower bound for $\mathcal{P}_d$, we need a $p$ that is "worst case" among all those who have the same value of $d$ and then use the corresponding lower bound for $\mathcal{P}_p$. In what follows, we start by giving a lower bound for $\mathcal{P}_p$, and then specialize it for $\mathcal{P}_d$.

We also assume that an oracle specifies the *true* probability of all observed categories. With this side-information, the best estimator *has* to use the true probabilities for the observed categories. For the unobserved categories, it needs to redistribute all the missing mass (the total probability of unobserved categories). Since the multiset of probabilities is fixed and any permutation of the remaining categories is equally probable, by symmetry there is no advantage in favoring one over the other. Therefore the best oracle-aided estimator is uniquely specified: exact probabilities for seen categories and uniform redistribution of the missing mass ($S_0$) over the unobserved categories. This argument can be proven formally via the maximin trick: substitute the maximum with a mean against an arbitrary prior over $p$, at which point the optimal $q$ is the posterior, and then optimize over priors. It then suffices to use the convexity of $p \log \frac{p}{q}$ with respect to $q$.

## B.1 Proof of Theorem 2

*Proof.* Without loss of generality assume that $p_1 \geq p_2 \geq p_3 \geq \ldots \geq p_k$. Let $\gamma > 1$, we have:

$$
r_n(\mathcal{P}_p) = \min_q \max_{p \in \mathcal{P}_p} \mathbb{E}\left[ \sum_{j=1}^{k} p_j \log \frac{p_j}{q_j} \right] \geq \mathbb{E}\left[ \sum_{j=D+1}^{k} p_j \log \frac{p_j}{\frac{\sum_{j=D+1}^{k} p_j}{k-D}} \right]
$$

$$
= \mathbb{E}\left[ \sum_{j=D+1}^{k} p_j \log \frac{p_j(k-D)}{n \sum_{j=D+1}^{k} p_j} \;\middle|\; D < \gamma d \right] \Pr\left(D < \gamma d\right)
$$

$$
+ \mathbb{E}\left[ \sum_{j=D+1}^{k} p_j \log \frac{p_j(k-D)}{\sum_{j=D+1}^{k} p_j} \;\middle|\; D \geq \gamma d \right] \Pr\left(D \geq \gamma d\right)
$$

$$
\overset{(a)}{\geq} \left(1 - \frac{1}{\gamma}\right) \sum_{j=\gamma d}^{k} p_j \log \frac{p_j(k - \gamma d)}{\sum_{j=\gamma d}^{k} p_j}
$$

where $(a)$ is by the following arguments: By Markov's inequality we have $\Pr\left(D \geq \gamma d\right) \leq \frac{1}{\gamma}$. Also, $\sum_{j=D+1}^{k} p_j \log \frac{np_j(k-D)}{n \sum_{j=D+1}^{k} p_j}$ is positive and decreasing in $D$ (in the extreme case, when $D = k$ is zero). Therefore,

$$
r_n(\mathcal{P}_p) \geq \left(1 - \frac{1}{\gamma}\right) \left( \sum_{j=\gamma d}^{k} p_j \right) \log \frac{k - \gamma d}{\sum_{j=\gamma d}^{k} p_j} + \sum_{i=\gamma d} p_j \log p_j
$$

This completes the proof. For any specific classes of distributions, we can find a lower bound by calculating $d$, $\sum_{j=\gamma d}^{k} p_j$, and $\sum_{j=\gamma d}^{k} p_j \log p_j$ for some $\gamma > 1$. $\qquad\square$

## B.2 Proof of Corollary 3

*Proof.* To use Theorem 2, we first calculate $d$, $\sum_{j>L} p_j$, and $\sum_{j>L} p_j \log p_j$ and then let $L = \gamma d$ for $\gamma = 2$.

$$
\sum_{j=L+1}^{k} p_j = \sum_{j=L+1}^{k} \frac{c}{j^\alpha}
$$

$$
\overset{(a)}{\geq} \int_{L+1}^{k+1} \frac{c}{x^\alpha} dx
$$

$$
= \frac{c}{\alpha - 1}\left[ (L+1)^{1-\alpha} - (k+1)^{1-\alpha} \right],
$$

where $(a)$ is by integration bound for monotone series. Similarly, we can show:

$$
\sum_{j=L+1}^{k} p_j \leq \int_{L}^{k} \frac{c}{x^\alpha} dx = \frac{c}{\alpha - 1}\left[ L^{1-\alpha} - k^{1-\alpha} \right].
$$

For the last summation in the lower bound of Theorem 2 we have:

$$\sum_{j=L+1}^{k} p_j \log p_j$$

$$= \sum_{j=L+1}^{k} \frac{c}{j^\alpha} \log \frac{c}{j^\alpha}$$

$$= c \sum_{j=L+1}^{k} \frac{1}{j^\alpha} \log \frac{1}{j^\alpha} + \log c \sum_{j=L+1}^{k} \frac{c}{j^\alpha}$$

$$\overset{(a)}{\geq} c \int_{L+1}^{k+1} \frac{1}{j^\alpha} \log \frac{1}{j^\alpha} dj + \log c \sum_{j=L+1}^{k} p_j$$

$$\overset{(b)}{=} \frac{c}{\alpha} \int_{(k+1)^{-\alpha}}^{(L+1)^{-\alpha}} x^{-\frac{1}{\alpha}} \log x \, dx + \log c \sum_{j=L+1}^{k} p_j$$

$$\geq \frac{c}{\alpha - 1} \left[ x^{1-\frac{1}{\alpha}} \log x \right]_{(k+1)^{-\alpha}}^{(L+1)^{-\alpha}} - \frac{c}{\alpha - 1} \int_{(k+1)^{-\alpha}}^{(L+1)^{-\alpha}} x^{-\frac{1}{\alpha}} + \frac{c \log c}{\alpha - 1} \left[ (L+1)^{1-\alpha} - (k+1)^{1-\alpha} \right]$$

Using Theorem 2, if $k > \max\{n, \left( \frac{10}{9} \right)^{\frac{1}{\alpha-1}} n^{\frac{1}{\alpha}}\}$ we have,

$$r_n(\mathcal{P})$$

$$\geq \frac{c}{\alpha - 1} (L+1)^{1-\alpha} \log \frac{k-2d}{\frac{c}{\alpha-1}(L+1)^{1-\alpha}} + (L+1)^{1-\alpha} \left[ \frac{c}{\alpha-1} \log(L+1)^{-\alpha} - \frac{c\alpha}{(\alpha-1)^2} + \frac{c \log c}{\alpha - 1} \right]$$

$$\geq \frac{c}{10(\alpha - 1)} (L+1)^{1-\alpha} \log \frac{k-2d}{(L+1)} + \frac{c}{\alpha-1}(L+1)^{1-\alpha} \left[ \frac{1}{10} \log(\alpha-1) - \frac{\alpha}{\alpha-1} \right]$$

where $(a)$ is by integration bound for monotone series, and $(b)$ is by change of variable $x = \frac{1}{j^\alpha}$.
Using Equation (3), choosing $L = 2d$,

$$r_n(\mathcal{P}) \geq \frac{2^{1-\alpha}c}{\alpha-1} d^{1-\alpha} \log \frac{k-2d}{2d} + \frac{2^{1-\alpha}c}{\alpha-1} d^{1-\alpha} \left[ \log(\alpha-1) - \frac{\alpha}{\alpha-1} \right],$$

and since for power-law distributions, $d$ grows proportionally to $n^{\frac{1}{\alpha}}$, we can write

$$r_n(\mathcal{P}) \geq c_1 \frac{d}{n} \log \frac{k-2d}{2d}(1 - o(1)),$$

for some constants $c_1$ and $c_2$. To compare this with the upper bound in the proof of Theorem 1, note that we always have $\mathbb{E}[\Phi_1] \leq d$, but for power law distributions both expressions grow proportionally to $n^{\frac{1}{\alpha}}$ and furthermore $\mathbb{E}[\Phi_1]/d$ converges to a constant, $\frac{1}{\alpha}$. This shows that the upper and lower bounds for power-law distributions are tight in the first order term, when $k$ is large. $\quad\square$

### B.3 proof of Corollary 4

*Proof.* To use Theorem 2, we first calculate $d$, $\sum_{j>\gamma d} p_j$, and $\sum_{j>\gamma d} p_j \log p_j$. For the expected number of distinct categories,

$$d = \sum_{j=1}^{k'} 1 - e^{-np_j} = k'(1 - e^{-\frac{n}{k'}}) = \frac{1-e^{-\rho}}{\rho} n.$$

For the sum of probabilities of unobserved categories,

$$\sum_{j>\gamma d} p_j = \frac{k' - \gamma d}{k'} = 1 - \frac{\gamma n (1 - e^{-\rho})}{\rho k'} = 1 - \gamma(1 - e^{-\rho}),$$

and for the last summation in (3),

$$\sum_{j=\gamma d+1}^{k} p_j \log p_j = \frac{k'-\gamma d}{k'} \log(\frac{1}{k'}) = \left(1 - \gamma(1 - e^{-\rho})\right) \log\left(\frac{\rho}{n}\right).$$

Therefore, by (3) we have $r_n(\mathcal{P}) \geq \left(1 - \frac{1}{\gamma}\right)\left(1 - \gamma(1 - e^{-\rho})\right) \log \frac{k-\gamma d}{1-\gamma(1-e^{-\rho})} + \left(1 - \gamma(1 - e^{-\rho})\right) \log\left(\frac{\rho}{n}\right)$, which can also be written as $r_n(\mathcal{P}) \geq \left(1 - \frac{1}{\gamma}\right) \frac{\rho\left(1-\gamma(1-e^{-\rho})\right)}{1-e^{-\rho}} \frac{d}{n} \log \frac{k-\gamma d}{d} + \left(1 - \gamma(1 - e^{-\rho})\right) \log \frac{1-e^{-\rho}}{(1-\gamma(1-e^{-\rho}))} + \frac{1}{\gamma}\left(1 - \gamma(1 - e^{-\rho})\right) \log \frac{1-e^{-\rho}}{d}$. Choosing $\gamma = 1.2$ and having $k > n^5$, the corollary follows for $\rho \leq 1.75$. $\qquad\square$

# C  Proofs of Good–Turing and absolute-discount relationship

## C.1  Proof of Lemma 5

*Proof.* For notational simplicity we define $C(\mu) := \frac{c^{\frac{1}{\alpha}}\Gamma(\mu-\frac{1}{\alpha})}{\alpha\mu!} n^{\frac{1}{\alpha}}$. Using Lemma 15,

$$\begin{aligned}
\frac{\mathbb{E}[\Phi_{\mu+1}]}{\mathbb{E}[\Phi_\mu]} &\overset{(a)}{\leq} \frac{C(\mu+1) + \mathcal{O}(\mu^{-\frac{1}{2}})}{C(\mu)\left(1 - \mathcal{O}(\mu^{-1}n^{-\frac{1}{\alpha}})\right) - \mathcal{O}(\mu^{-\frac{1}{2}})} \\
&\leq \frac{C(\mu+1)}{C(\mu)}\left(\frac{1 + \mathcal{O}(\mu^{-\frac{1}{2}}C^{-1}(\mu+1))}{1 - \mathcal{O}(\mu^{-1}n^{-\frac{1}{\alpha}}) - \mathcal{O}(\mu^{-\frac{1}{2}}C^{-1}(\mu))}\right) \\
&\leq \frac{C(\mu+1)}{C(\mu)}\left(1 + \mathcal{O}(\mu^{-\frac{1}{2}}C^{-1}(\mu+1)) + \mathcal{O}(\mu^{-1}n^{-\frac{1}{\alpha}})\right) \\
&\overset{(b)}{\leq} \frac{C(\mu+1)}{C(\mu)}\left(1 + \mathcal{O}(\mu^{-\frac{1}{2}+1+\frac{1}{\alpha}}n^{\frac{-1}{\alpha}})\right) \\
&\overset{(c)}{=} \frac{\mu - \frac{1}{\alpha}}{\mu+1}\left(1 + \mathcal{O}(n^{\frac{-3}{2(2\alpha+1)}})\right)
\end{aligned}$$

The inequality in $(a)$ and $(b)$ are by Lemma 15 and $(c)$ is by the fact that $\mu < n^{\frac{1}{2\alpha+1}}$. $\qquad\square$

## C.2  Proof of Theorem 6

Recall that $S_\mu$ denotes the total probability of symbols appearing $\mu$ times, and let $\hat{S}_\mu$ be the probability assigned to those symbols by an estimator. Note that, given the samples, we may think of $S$ and $\hat{S}$ as legitimate probability distributions on the set $\mu = 0, 1, \cdots, n$. In [OS15], it was shown that the competitive loss of an estimator over a class defined by a single distribution $p$ and its permutations can be bounded by:

$$\epsilon_n(\mathcal{P}_p, q) = r_n(p, q) - r_n(\mathcal{P}_p) \leq \mathbb{E}[\mathsf{KL}(S||\hat{S})].$$

This is well defined, since $S$ and $\hat{S}$ only refer to the multiset probabilities, which stays invariant over all distributions in the class. Using this bound and the equivalence of Lemma 5, we can proceed with the proof. In the proof, we analyze the absolute-discount estimator with discount $\delta = \min\{\frac{\max\{\Phi_1, 1\}}{D}, \delta_{\max}\}$.

*Proof.* We have:

$$\mathsf{KL}(S||\hat{S})$$

$$= \sum_{\mu=0}^{\infty} S_\mu \log \frac{S_\mu}{\hat{S}_\mu}$$

$$\overset{(a)}{\leq} \sum_{\mu=0}^{\infty} \frac{(S_\mu - \hat{S}_\mu)^2}{\hat{S}_\mu}$$

$$= \frac{(S_0 - \hat{S}_0)^2}{\hat{S}_0} + \sum_{\mu=1}^{\mu_0} \frac{(S_\mu - \hat{S}_\mu)^2}{\hat{S}_\mu} + \sum_{\mu=\mu_0+1}^{\infty} \frac{(S_\mu - \hat{S}_\mu)^2}{\hat{S}_\mu}$$

$$= \frac{(S_0 - \frac{D\delta}{n})^2}{\frac{D\delta}{n}} + \sum_{\mu=1}^{\mu_0} \frac{(S_\mu - \frac{\mu - \frac{1}{\alpha}}{n}\Phi_\mu + \frac{\mu-\frac{1}{\alpha}}{n}\Phi_\mu - \frac{\mu-\delta}{n}\Phi_\mu)^2}{\frac{\mu-\delta}{n}\Phi_\mu} + \sum_{\mu=\mu_0+1}^{\infty} \frac{(S_\mu - \frac{\mu-\delta}{n}\Phi_\mu)^2}{\frac{\mu-\delta}{n}\Phi_\mu}$$

$$\overset{(b)}{\leq} \frac{(S_0 - \frac{D\delta}{n})^2}{\frac{D\delta}{n}} + 2\sum_{\mu=1}^{\mu_0} \frac{(S_\mu - \frac{\mu-\frac{1}{\alpha}}{n}\Phi_\mu)^2}{\frac{\mu-\delta}{n}\Phi_\mu} + 2\sum_{\mu=1}^{\mu_0} \frac{(\frac{\mu-\frac{1}{\alpha}}{n} - \frac{\mu-\delta}{n})^2 \Phi_\mu^2}{\frac{\mu-\delta}{n}\Phi_\mu} + \sum_{\mu=\mu_0+1}^{\infty} \frac{(S_\mu - \frac{\mu-\delta}{n}\Phi_\mu)^2}{\frac{\mu-\delta}{n}\Phi_\mu}$$

$$\tag{12}$$

where $(a)$ is by Lemma 14 and $(b)$ is by $(a+b)^2 \leq 2a^2 + 2b^2$. We choose $\mu_0 = n^{\frac{1}{2\alpha+1}}$ and show the proof for the case when $n^{\frac{1}{2\alpha+1}} \geq 20\log n$, namely, $\alpha \leq \frac{\log n}{2(\log\log n + \log 20)} - \frac{1}{2}$. For $\alpha > \frac{\log n}{2(\log\log n + \log 20)} - \frac{1}{2}$, the proof follows the same lines, but by a different choice of $\mu_0$. Lemmas 10, 11, 12, and 13 bound each term in Equation (12) separately, and hence

$$\mathbb{E}[\mathsf{KL}(S||\hat{S})] = \mathcal{O}\left(\frac{1}{n^{\frac{2\alpha-1}{2\alpha+1}}}\right). \qquad \square$$

**Lemma 10.** *For a power-law distribution with exponent $\alpha > \alpha_0 > 1$, and the choice of $\delta = \min\{\frac{\max\{\Phi_1, 1\}}{D}, \delta_{\max}\}$,*

$$\mathbb{E}\left[\frac{(S_0 - \frac{D\delta}{n})^2}{\frac{D\delta}{n}}\right] = \mathcal{O}\left(\frac{1}{n}\right).$$

*Proof.* To upper bound the first term of the KL loss in Equation (12), namely the loss of proposed estimator for the missing mass, let $A$ be the event $(1-t)\mathbb{E}[\Phi_1] \leq \Phi_1 \leq (1+t)\mathbb{E}[\Phi_1]$ and $\frac{1-t}{1+t}\frac{\mathbb{E}[\Phi_1]}{d} \leq$

$\frac{\Phi_1}{D} \leq \frac{1+t}{1-t}\frac{\mathbb{E}[\Phi_1]}{d}$ for some $0 < t < 1$,

$$\mathbb{E}\left[\frac{(S_0 - \frac{D\delta}{n})^2}{\frac{D\delta}{n}}\right] = \mathbb{E}\left[\frac{(S_0 - \frac{D\delta}{n})^2}{\frac{D\delta}{n}}\ \middle|\ A\right]\Pr(A) + \mathbb{E}\left[\frac{(S_0 - \frac{D\delta}{n})^2}{\frac{D\delta}{n}}\ \middle|\ A^c\right]\Pr(A^c)$$

$$\overset{(a)}{\leq} \mathbb{E}\left[\frac{(S_0 - \frac{D\delta}{n})^2}{\frac{D\delta}{n}}\ \middle|\ A\right]\Pr(A) + 4\exp\left(-\frac{t^2\mathbb{E}[\Phi_1]}{2(1+t/3)}\right)n^2$$

$$\overset{(b)}{\leq} \mathbb{E}\left[\frac{2(S_0 - \frac{\mathbb{E}[\Phi_1]}{n})^2 + 2(\frac{\mathbb{E}[\Phi_1]}{n} - \frac{\Phi_1}{n})^2}{\frac{\Phi_1}{n}}\ \middle|\ A\right]\Pr(A) + 4n^2\exp\left(-\frac{t^2\mathbb{E}[\Phi_1]}{2(1+t/3)}\right)$$

$$\overset{(c)}{\leq} \frac{\mathbb{E}\left[2(S_0 - \frac{\mathbb{E}[\Phi_1]}{n})^2 + 2(\frac{\mathbb{E}[\Phi_1]}{n} - \frac{\Phi_1}{n})^2\ \middle|\ A\right]\Pr(A)}{\frac{(1-t)\mathbb{E}[\Phi_1]}{n}} + 4n^2\exp\left(-\frac{t^2\mathbb{E}[\Phi_1]}{2(1+t/3)}\right)$$

$$\overset{(d)}{\leq} \frac{2\mathrm{Var}(S_0) + \frac{2}{n^2}\mathrm{Var}(\Phi_1)}{\frac{\mathbb{E}[\Phi_1]}{2n}} + o\left(\frac{1}{n}\right)$$

$$\overset{(e)}{\leq} \frac{\frac{4}{n^2}\mathbb{E}[\Phi_2] + \frac{2}{n^2}\mathbb{E}[\Phi_1]}{\frac{\mathbb{E}[\Phi_1]}{2n}} + o\left(\frac{1}{n}\right)$$

$$= \mathcal{O}\left(\frac{1}{n}\right),$$

where $(b)$ is by choosing $t$ such that $\frac{1+t}{1-t}\frac{1}{\alpha_0} < \delta_{\max}$ and therefore conditioned on $A$, $\delta = \frac{\Phi_1}{D}$. Also, $(c)$ is by concentration of $\Phi_1$ (see Lemma 16), $(d)$ is by choosing $t = n^{-\frac{1}{4\alpha}}$, and $(e)$ is because $\mathrm{Var}(\Phi_1) \leq \mathbb{E}[\Phi_1]$ and $\mathrm{Var}(S_0) \leq \frac{2}{n^2}\mathbb{E}[\Phi_2]$ (see Lemma 21). $\qquad\square$

**Lemma 11.** *For a power-law distribution with exponent $\alpha$ and choice of $\mu_0 = \mathcal{O}(n^{\frac{1}{2\alpha+1}})$,*

$$\mathbb{E}\left[\sum_{\mu=1}^{\mu_0}\frac{(S_\mu - \frac{\mu - \frac{1}{\alpha}}{n}\Phi_\mu)^2}{\frac{\mu-\delta}{n}\Phi_\mu}\right] = \mathcal{O}\left(n^{\frac{1-2\alpha}{2\alpha+1}}\right)$$

*Proof.* Using Lemma 5 and $(a+b)^2 \leq 2a^2 + 2b^2$, we bound the second term in (12):

$$\mathbb{E}\left[\sum_{\mu=1}^{\mu_0}\frac{(S_\mu - \frac{\mu - \frac{1}{\alpha}}{n}\Phi_\mu)^2}{\frac{\mu-\delta}{n}\Phi_\mu}\right]$$

$$\leq 2\mathbb{E}\left[\sum_{\mu=1}^{\mu_0}\frac{(\frac{\mu+1}{n}\frac{\mathbb{E}[\Phi_{\mu+1}]}{\mathbb{E}[\Phi_\mu]}\Phi_\mu\mathcal{O}(n^{\frac{-3}{2(2\alpha+1)}}))^2}{\frac{\mu-\delta}{n}\Phi_\mu}\right] + 2\mathbb{E}\left[\sum_{\mu=1}^{\mu_0}\frac{(S_\mu - \frac{\mu+1}{n}\frac{\mathbb{E}[\Phi_{\mu+1}]}{\mathbb{E}[\Phi_\mu]}\Phi_\mu)^2}{\frac{\mu-\delta}{n}\Phi_\mu}\right] \qquad (13)$$

For the first term in right hand side of Equation (13),

$$\mathbb{E}\left[\sum_{\mu=1}^{\mu_0}\frac{(\frac{\mu+1}{n}\frac{\mathbb{E}[\Phi_{\mu+1}]}{\mathbb{E}[\Phi_\mu]}\Phi_\mu\mathcal{O}(n^{\frac{-3}{2(2\alpha+1)}}))^2}{\frac{\mu-\delta}{n}\Phi_\mu}\right] \leq n^{-1-\frac{3}{2\alpha+1}}\sum_{\mu=1}^{\mu_0}\frac{(\mu+1)^2\left(\frac{\mathbb{E}[\Phi_{\mu+1}]}{\mathbb{E}[\Phi_\mu]}\right)^2\mathbb{E}[\Phi_\mu]}{\mu - \delta}$$

$$\leq \frac{4}{1-\delta_{\max}}n^{\frac{1}{\alpha}-1-\frac{3}{2\alpha+1}}\sum_{\mu=1}^{\mu_0}\mu^{-\frac{1}{\alpha}}$$

$$\leq \frac{4}{1-\delta_{\max}}n^{\frac{1}{\alpha}-1-\frac{3}{2\alpha+1}}\mu_0^{1-\frac{1}{\alpha}} = \mathcal{O}\left(\frac{1}{n}\right),$$

where the last line is by choosing $\mu_0 = n^{\frac{1}{2\alpha+1}}$. For the second term in Equation 13, using $(a+b)^2 \leq 2a^2 + 2b^2$ we have,

$$\left(S_\mu - \frac{\mu+1}{n}\frac{\mathbb{E}[\Phi_{\mu+1}]}{\mathbb{E}[\Phi_\mu]}\Phi_\mu\right)^2 = \left[S_\mu - \frac{\mu+1}{n}\mathbb{E}[\Phi_{\mu+1}] + \frac{\mu+1}{n}\mathbb{E}[\Phi_{\mu+1}] - \frac{\mu+1}{n}\frac{\mathbb{E}[\Phi_{\mu+1}]}{\mathbb{E}[\Phi_\mu]}\Phi_\mu\right]^2$$

$$\leq 2\left(S_\mu - \frac{\mu+1}{n}\mathbb{E}[\Phi_{\mu+1}]\right)^2 + 2\left(\frac{\mu+1}{n}\frac{\mathbb{E}[\Phi_{\mu+1}]}{\mathbb{E}[\Phi_\mu]}\Phi_\mu - \frac{\mu+1}{n}\mathbb{E}[\Phi_{\mu+1}]\right)^2,$$

and therefore:

$$\mathbb{E}\left[\sum_{\mu=1}^{\mu_0} \frac{(S_\mu - \frac{\mu+1}{n}\frac{\mathbb{E}[\Phi_{\mu+1}]}{\mathbb{E}[\Phi_\mu]}\Phi_\mu)^2}{\frac{\mu-\delta}{n}\Phi_\mu}\right]$$

$$= \mathbb{E}\left[\sum_{\mu=1}^{\mu_0} \frac{(S_\mu - \frac{\mu+1}{n}\frac{\mathbb{E}[\Phi_{\mu+1}]}{\mathbb{E}[\Phi_\mu]}\Phi_\mu)^2}{\frac{\mu-\delta}{n}\Phi_\mu} \,\Big|\, \Phi_\mu \geq \frac{\mathbb{E}[\Phi_\mu]}{2}\right]\Pr\left(\Phi_\mu \geq \frac{\mathbb{E}[\Phi_\mu]}{2}\right) +$$

$$\mathbb{E}\left[\sum_{\mu=1}^{\mu_0} \frac{(S_\mu - \frac{\mu+1}{n}\frac{\mathbb{E}[\Phi_{\mu+1}]}{\mathbb{E}[\Phi_\mu]}\Phi_\mu)^2}{\frac{\mu-\delta}{n}\Phi_\mu} \,\Big|\, \Phi_\mu < \frac{\mathbb{E}[\Phi_\mu]}{2}\right]\Pr\left(\Phi_\mu < \frac{\mathbb{E}[\Phi_\mu]}{2}\right)$$

$$\stackrel{(a)}{\leq} \mathbb{E}\left[\sum_{\mu=1}^{\mu_0} \frac{(S_\mu - \frac{\mu+1}{n}\frac{\mathbb{E}[\Phi_{\mu+1}]}{\mathbb{E}[\Phi_\mu]}\Phi_\mu)^2}{\frac{\mu-\delta}{n}\Phi_\mu} \,\Big|\, \Phi_\mu \geq \frac{\mathbb{E}[\Phi_\mu]}{2}\right]\Pr\left(\Phi_\mu \geq \frac{\mathbb{E}[\Phi_\mu]}{2}\right) + n^2\exp\left(-\frac{1}{6\mu_0}\left(\frac{n}{\mu_0}\right)^{\frac{1}{\alpha}}\right)$$

$$\leq \left(\sum_{\mu=1}^{\mu_0} \frac{\mathbb{E}[(S_\mu - \frac{\mu+1}{n}\frac{\mathbb{E}[\Phi_{\mu+1}]}{\mathbb{E}[\Phi_\mu]}\Phi_\mu)^2]}{\frac{\mu-\delta}{2n}\mathbb{E}[\Phi_\mu]} \,\Big|\, \Phi_\mu \geq \frac{\mathbb{E}[\Phi_\mu]}{2}\right)\Pr\left(\Phi_\mu \geq \frac{\mathbb{E}[\Phi_\mu]}{2}\right) + n^2\exp\left(-\frac{1}{6\mu_0}\left(\frac{n}{\mu_0}\right)^{\frac{1}{\alpha}}\right)$$

$$\stackrel{(b)}{\leq} \sum_{\mu=1}^{\mu_0} \frac{\mathbb{E}\left[2\left(S_\mu - \frac{\mu+1}{n}\mathbb{E}[\Phi_{\mu+1}]\right)^2 + 2\left(\frac{\mu+1}{n}\frac{\mathbb{E}[\Phi_{\mu+1}]}{\mathbb{E}[\Phi_\mu]}\Phi_\mu - \frac{\mu+1}{n}\mathbb{E}[\Phi_{\mu+1}]\right)^2\right]}{\frac{\mu-\delta}{2n}\mathbb{E}[\Phi_\mu]} + n^2\exp\left(-\frac{1}{6\mu_0}\left(\frac{n}{\mu_0}\right)^{\frac{1}{\alpha}}\right)$$

$$\leq \sum_{\mu=1}^{\mu_0} \frac{2\mathrm{Var}(S_\mu) + 2\left(\frac{\mu+1}{n}\frac{\mathbb{E}[\Phi_{\mu+1}]}{\mathbb{E}[\Phi_\mu]}\right)^2\mathrm{Var}(\Phi_\mu)}{\frac{\mu-\delta_{\max}}{2n}\mathbb{E}[\Phi_\mu]} + n^2\exp\left(-\frac{1}{6\mu_0}\left(\frac{n}{\mu_0}\right)^{\frac{1}{\alpha}}\right)$$

$$\stackrel{(d)}{\leq} \sum_{\mu=1}^{\mu_0} \frac{2\frac{(\mu+2)^2}{n^2}\mathbb{E}[\Phi_{\mu+2}] + 2\frac{(\mu+1)^2}{n^2}\frac{\mathbb{E}^2[\Phi_{\mu+1}]}{\mathbb{E}[\Phi_\mu]}}{\frac{\mu-\delta_{\max}}{2n}\mathbb{E}[\Phi_\mu]} + n^2\exp\left(-\frac{1}{6\mu_0}\left(\frac{n}{\mu_0}\right)^{\frac{1}{\alpha}}\right)$$

$$\stackrel{(e)}{\leq} \sum_{\mu=1}^{\mu_0} \frac{3}{1-\delta_{\max}}\frac{\mu}{n} + o\left(\frac{1}{n}\right)$$

$$\stackrel{(f)}{\leq} \frac{3}{n(1-\delta_{\max})}\left(\frac{\mu_0^2}{2} + 2\mu_0\right) + o\left(\frac{1}{n}\right) = \mathcal{O}\left(n^{\frac{1-2\alpha}{2\alpha+1}}\right).$$

Note that $(a)$ follows from Lemma 16, $(b)$ from $(x+y)^2 \leq 2x^2 + 2y^2$, and $(c)$ from $\mathbb{E}[S_\mu] = \frac{\mu+1}{n}\mathbb{E}[\Phi_{\mu+1}]$ and $\delta < \delta_{\max}$. Also, $(d)$ results from $\mathrm{Var}[\Phi_\mu] \leq \mathbb{E}[\Phi_\mu]$ and $\mathrm{Var}[S_\mu] \leq$

$\frac{(\mu+2)^2}{n^2}\mathbb{E}[\Phi_{\mu+2}]$ (see Lemma 21), $(e)$ is by Lemma 15, and $(f)$ results from the choice of $\mu_0 = n^{\frac{1}{2\alpha+1}}$.  $\square$

**Lemma 12.** *For a power-law distribution with exponent $\alpha$ and the choice of $\mu_0 = n^{\frac{1}{2\alpha+1}}$,*

$$\mathbb{E}\left[\sum_{\mu=1}^{\mu_0}\frac{\left(\frac{\mu-\frac{1}{\alpha}}{n}-\frac{\mu-\delta}{n}\right)^2\Phi_\mu^2}{\frac{\mu-\delta}{n}\Phi_\mu}\right] = \mathcal{O}\left(\frac{n^{\frac{1}{2\alpha}}}{n}\right).$$

*Proof.*

$$\mathbb{E}\left[\sum_{\mu=1}^{\mu_0}\frac{\left(\frac{\mu-\frac{1}{\alpha}}{n}-\frac{\mu-\delta}{n}\right)^2\Phi_\mu^2}{\frac{\mu-\delta}{n}\Phi_\mu}\right] \leq \frac{1}{n}\sum_{\mu=1}^{\mu_0}\frac{\mathbb{E}\left[(\frac{1}{\alpha}-\delta)^2\Phi_\mu\right]}{\mu-\delta_{\max}}.$$

Similar to the proof of Lemma 10, let $A$ be the event $(1-t)\mathbb{E}[\Phi_1] \leq \Phi_1 \leq (1+t)\mathbb{E}[\Phi_1]$ and $\frac{1-t}{1+t}\frac{\mathbb{E}[\Phi_1]}{d} \leq \frac{\Phi_1}{D} \leq \frac{1+t}{1-t}\frac{\mathbb{E}[\Phi_1]}{d}$ for some $0 < t < 1$. Thus,

$$\begin{aligned}
\mathbb{E}\left[(\frac{1}{\alpha}-\delta)^2\Phi_\mu\right] &= \mathbb{E}\left[(\frac{1}{\alpha}-\delta)^2\Phi_\mu \,\Big|\, A\right]\Pr(A) + \mathbb{E}\left[(\frac{1}{\alpha}-\delta)^2\Phi_\mu \,\Big|\, A^c\right]\Pr(A^c)\\
&\leq t^2\Pr(A)\,\mathbb{E}\left[\Phi_\mu \,\Big|\, A\right] + \Pr(A^c)\,\mathbb{E}\left[\Phi_\mu \,\Big|\, A^c\right]\\
&\leq n^{-\frac{1}{2\alpha}}\mathbb{E}\left[\Phi_\mu\right]
\end{aligned}$$

where the last line is by choosing $t = n^{-\frac{1}{4\alpha}}$ and using Lemma 22. Hence, we have

$$\mathbb{E}\left[\sum_{\mu=1}^{\mu_0}\frac{\left(\frac{\mu-\frac{1}{\alpha}}{n}-\frac{\mu-\delta}{n}\right)^2\Phi_\mu^2}{\frac{\mu-\delta}{n}\Phi_\mu}\right] \leq \frac{n^{-\frac{1}{2\alpha}}}{n}\sum_{\mu=1}^{\mu_0}\frac{\mathbb{E}\left[\Phi_\mu\right]}{\mu-\delta_{\max}} = \mathcal{O}\left(\frac{n^{\frac{1}{2\alpha}}}{n}\right),$$

where the constant depends on $\delta_{\max}$ and therefore on $\alpha_0$.  $\square$

**Lemma 13.** *For a power-law distribution with exponent $\alpha$, and $\mu_0 = n^{\frac{1-2\alpha}{2\alpha+1}}$,*

$$\mathbb{E}\left[\sum_{\mu=\mu_0+1}^{\infty}\frac{(S_\mu-\frac{\mu-\delta}{n}\Phi_\mu)^2}{\frac{\mu-\delta}{n}\Phi_\mu}\right] = \mathcal{O}\left(n^{\frac{1-2\alpha}{2\alpha+1}}\right)$$

*Proof.* For the last part in Equation 12:

$$\mathbb{E}\left[\sum_{\mu=\mu_0+1}^{\infty}\frac{(S_\mu-\frac{\mu-\delta}{n}\Phi_\mu)^2}{\frac{\mu-\delta}{n}\Phi_\mu}\right] \leq \mathbb{E}\left[\sum_{\mu=\mu_0+1}^{\infty}\frac{2(S_\mu-\frac{\mu}{n}\Phi_\mu)^2 + 2(\frac{\delta\Phi_\mu}{n})^2}{\frac{\mu-\delta}{n}\Phi_\mu}\right].$$

We bound both terms in the above expression separately. For the second term, we have:

$$\mathbb{E}\left[\sum_{\mu=\mu_0+1}^{\infty}\frac{(\frac{\delta\Phi_\mu}{n})^2}{\frac{\mu-\delta}{n}\Phi_\mu}\right] \leq \frac{1}{n}\sum_{\mu=\mu_0+1}^{\infty}\frac{\mathbb{E}[\Phi_\mu]}{\mu-\delta_{\max}} \leq \frac{2c^{\frac{1}{\alpha}}\Gamma\left(1-\frac{1}{\alpha}\right)}{\alpha n}\frac{1}{\mu_0}\left(\frac{n}{\mu_0}\right)^{\frac{1}{\alpha}} + \frac{2}{n\sqrt{\mu_0}} = \mathcal{O}\left(n^{-\frac{2\alpha}{2\alpha+1}}\right),$$

and for the first part, we have:

$$\sum_{\mu=\mu_0+1}^{\infty} \frac{(S_\mu - \frac{\mu}{n}\Phi_\mu)^2}{\frac{\mu-\delta}{n}\Phi_\mu}$$

$$\overset{(a)}{\leq} \sum_{\mu=\mu_0+1}^{\infty} \sum_{x} \mathbb{1}_x^\mu \frac{(p_x - \frac{\mu}{n})^2}{\frac{\mu-1}{n}}$$

$$\leq 2 \sum_{x:\, np_x \geq \frac{\mu_0}{2}} \sum_{\mu=\mu_0+1}^{\infty} \mathbb{1}_x^\mu \frac{(p_x - \frac{\mu}{n})^2}{\frac{\mu}{n}} + 2 \sum_{x:\, np_x < \frac{\mu_0}{2}} \sum_{\mu=\mu_0+1}^{\infty} \mathbb{1}_x^\mu \frac{(p_x - \frac{\mu}{n})^2}{\frac{\mu}{n}}$$

$$\leq 2 \sum_{x:\, np_x \geq \frac{\mu_0}{2}} \sum_{\mu=1}^{\infty} \mathbb{1}_x^\mu \frac{(p_x - \frac{\mu}{n})^2}{\frac{\mu}{n}} + 2 \sum_{x:\, np_x < \frac{\mu_0}{2}} \frac{n}{\mu_0} \mathbb{1}_x^{>\mu_0},$$

where $(a)$ follows from $(\sum_{i=1}^n a_i)^2 \leq n(\sum_{i=1}^n a_i^2)$. Taking expectations of both sides:

$$\mathbb{E}\left[ \sum_{\mu=\mu_0+1}^{\infty} \frac{(S_\mu - \frac{\mu}{n}\Phi_\mu)^2}{\frac{\mu-\delta}{n}\Phi_\mu} \right] \overset{(a)}{\leq} 2\left(\frac{2nc}{\mu_0}\right)^{\frac{1}{\alpha}} \frac{1}{n} \mathbb{E}\left[ \frac{(np_x)^2 - 2\mu np_x + \mu^2}{\mu} \right] + 2 \sum_{x:\, np_x < \frac{\mu_0}{2}} \frac{n}{\mu_0} \mathbb{E}[\mathbb{1}_x^{>\mu_0}]$$

$$\overset{(b)}{\leq} 2\left(\frac{2nc}{\mu_0}\right)^{\frac{1}{\alpha}} \frac{3}{n} + 2 \sum_{x:\, np_x < \frac{\mu_0}{2}} \frac{n}{\mu_0} \mathbb{E}[\mathbb{1}_x^{>\mu_0}]$$

$$\overset{(c)}{\leq} \left(\frac{2nc}{\mu_0}\right)^{\frac{1}{\alpha}} \frac{6}{n} + 2 \sum_{x:\, np_x < \frac{\mu_0}{2}} \frac{n}{\mu_0} \exp\left(\mu_0 - np_x - \mu_0 \ln\left(\frac{\mu_0}{np_x}\right)\right)$$

$$\overset{(d)}{\leq} \left(\frac{2nc}{\mu_0}\right)^{\frac{1}{\alpha}} \frac{6}{n} + 2 \sum_{x:\, np_x < \frac{\mu_0}{2}} \frac{n}{\mu_0} \exp\left(\frac{np_x - \mu_0}{3}\right)$$

$$\overset{(e)}{\leq} \left(\frac{2nc}{\mu_0}\right)^{\frac{1}{\alpha}} \frac{4}{n} + 2e^{-\frac{\mu_0}{6}}\left(\frac{n}{\mu_0}\right)^2$$

$$= \mathcal{O}(n^{\frac{1-2\alpha}{2\alpha+1}}),$$

where $(a)$ is by bounding the number of elements with probability greater than $\mu_0/2n$, $(b)$ follows from the fact that $\mathbb{E}[\frac{1}{\mu}]$ when $\mu$ is a Poisson distribution with mean $\lambda$, is bounded by $\frac{1}{\lambda} + \frac{3}{\lambda^2}$ (note that $\mu = 0$ is excluded). Also, $(c)$ follows from Lemma 18, $(d)$ follows from $3(x - 1 - x \ln x) \leq 1 - x$ for $x > 2$, and $(e)$ is by convexity of the exponential term in $p_x$ and the fact that a convex function is maximized at the boundaries. $\qquad\square$

# D  Tools

This section provides a summary of tools used in the proofs throughout the paper.

**Lemma 14.** *For two distributions $p$ and $q$,*

$$\mathsf{KL}(p||q) := \sum_i p_i \log \frac{p_i}{q_i} \leq \sum_i \frac{(p_i - q_i)^2}{q_i}$$

**Lemma 15.** *For a power-law distribution with power $\alpha > \alpha_0 > 1$ and normalization factor $c$, for $\mu \geq 1$*

$$\mathbb{E}[\Phi_\mu] \leq \frac{c^{\frac{1}{\alpha}} \Gamma\left(\mu - \frac{1}{\alpha}\right)}{\alpha \mu!} n^{\frac{1}{\alpha}} + \frac{1}{\sqrt{2\pi\mu}} \leq \frac{c^{\frac{1}{\alpha}} \Gamma(1 - \frac{1}{\alpha})}{\mu\alpha} \left(\frac{n}{\mu}\right)^{\frac{1}{\alpha}} + \frac{1}{\sqrt{2\pi\mu}}.$$

*Also, for $1 \leq \mu < n^{\frac{1}{\alpha+1}}$, and $k > n^{\frac{1}{\alpha-1}}$,*

$$\mathbb{E}[\Phi_\mu] \geq \frac{c^{\frac{1}{\alpha}} \Gamma\left(\mu - \frac{1}{\alpha}\right)}{\alpha \mu!} n^{\frac{1}{\alpha}} - \frac{1}{\sqrt{2\pi\mu}}.$$

*Proof.* For the upper bound on the expected number of elements that appeared $\mu$ times:

$$\mathbb{E}[\Phi_\mu] = \mathbb{E}\left[\sum_{x=1}^{k} \mathbb{1}_x^\mu\right]$$

$$= \sum_{x=1}^{k} e^{-np_x} \frac{(np_x)^\mu}{\mu!}$$

$$= \sum_{x=1}^{k} e^{-\frac{nc}{x^\alpha}} \frac{\left(\frac{nc}{x^\alpha}\right)^\mu}{\mu!}$$

$$\overset{(a)}{\leq} \int_1^k e^{-\frac{nc}{x^\alpha}} \frac{\left(\frac{nc}{x^\alpha}\right)^\mu}{\mu!} dx + \max_x \left\{ e^{-\frac{nc}{x^\alpha}} \frac{\left(\frac{nc}{x^\alpha}\right)^\mu}{\mu!} \right\}$$

$$\overset{(b)}{=} \frac{(nc)^{\frac{1}{\alpha}}}{\alpha\mu!} \int_{\frac{nc}{k^\alpha}}^{nc} e^{-y} y^{\mu-1-\frac{1}{\alpha}} dy + \max_x \left\{ e^{-\frac{nc}{x^\alpha}} \frac{\left(\frac{nc}{x^\alpha}\right)^\mu}{\mu!} \right\}$$

$$\overset{(c)}{\leq} \frac{(nc)^{\frac{1}{\alpha}}}{\alpha\mu!} \left[ \Gamma\left(\mu - \frac{1}{\alpha}, \frac{nc}{k^\alpha}\right) - \Gamma\left(\mu - \frac{1}{\alpha}, nc\right) \right] + \frac{1}{\sqrt{2\pi\mu}}$$

$$= \frac{c^{\frac{1}{\alpha}} \Gamma\left(\mu - \frac{1}{\alpha}\right)}{\alpha\mu!} n^{\frac{1}{\alpha}} + \frac{1}{\sqrt{2\pi\mu}}, \tag{14}$$

where $(a)$ is followed by the integration bound for a uni-modal series, $(b)$ is by changing of variables $\frac{nc}{x^\alpha} = y$. Also $(c)$ is by the definition of Gamma function and the fact that $e^{-t}t^\mu$ is maximized at $t = \mu$ followed by Stirling's approximation. By further simplifying the Gamma function term:

$$\frac{\Gamma(\mu - \frac{1}{\alpha})}{\mu!} = \frac{(\mu - 1 - \frac{1}{\alpha})(\mu - 2 - \frac{1}{\alpha})\ldots(1 - \frac{1}{\alpha})\Gamma(1 - \frac{1}{\alpha})}{\mu!}$$

$$= \frac{1}{\mu} \prod_{j=1}^{\mu-1} \left(1 - \frac{1}{j\alpha}\right) \Gamma\left(1 - \frac{1}{\alpha}\right)$$

$$= \frac{1}{\mu} \exp\left(\sum_{j=1}^{\mu-1} \log\left(1 - \frac{1}{j\alpha}\right)\right) \Gamma\left(1 - \frac{1}{\alpha}\right)$$

$$\overset{(a)}{\leq} \frac{1}{\mu} \exp\left(-\frac{1}{\alpha} \sum_{j=1}^{\mu-1} \frac{1}{j}\right) \Gamma\left(1 - \frac{1}{\alpha}\right)$$

$$\overset{(b)}{\leq} \frac{1}{\mu} \mu^{-\frac{1}{\alpha}} \Gamma\left(1 - \frac{1}{\alpha}\right).$$

where $(a)$ is by $\log(1-x) \leq -x$ for $0 < x < 1$, and $(b)$ is because $\sum_{j=1}^{t} \frac{1}{j} \geq \log(t+1)$. Similarly for the lower bound we have:

$$
\begin{aligned}
\mathbb{E}[\Phi_\mu] &= \mathbb{E}\left[\sum_{x=1}^{k} \mathbb{1}_x^\mu\right] \\
&= \sum_{x=1}^{k} e^{-np_x} \frac{(np_x)^\mu}{\mu!} \\
&= \sum_{x=1}^{k} e^{-\frac{nc}{x^\alpha}} \frac{\left(\frac{nc}{x^\alpha}\right)^\mu}{\mu!} \\
&\stackrel{(a)}{\geq} \int_1^k e^{-\frac{nc}{x^\alpha}} \frac{\left(\frac{nc}{x^\alpha}\right)^\mu}{\mu!} dx - \max_x\{e^{-\frac{nc}{x^\alpha}} \frac{\left(\frac{nc}{x^\alpha}\right)^\mu}{\mu!}\} \\
&\stackrel{(b)}{=} \frac{(nc)^{\frac{1}{\alpha}}}{\alpha\mu!} \int_{\frac{nc}{k^\alpha}}^{nc} e^{-y} y^{\mu-1-\frac{1}{\alpha}} dy - \frac{1}{\sqrt{2\pi\mu}} \\
&\stackrel{(c)}{=} \frac{(nc)^{\frac{1}{\alpha}}}{\alpha\mu!} \left[\Gamma\left(\mu - \frac{1}{\alpha}\right) - \gamma\left(\mu - \frac{1}{\alpha}, \frac{nc}{k^\alpha}\right) - \Gamma\left(\mu - \frac{1}{\alpha}, nc\right)\right] - \frac{1}{\sqrt{2\pi\mu}} \\
&\stackrel{(d)}{\geq} \frac{c^{\frac{1}{\alpha}}\Gamma\left(\mu - \frac{1}{\alpha}\right)}{\alpha\mu!} n^{\frac{1}{\alpha}} \left(1 - \mathcal{O}\left(\mu^{-1} n^{-\frac{1}{\alpha}}\right)\right) - \frac{1}{\sqrt{2\pi\mu}},
\end{aligned}
\tag{15}
$$

By Lemma 20 we have $\gamma\left(\mu - \frac{1}{\alpha}, \frac{nc}{k^\alpha}\right) \leq \frac{1}{\mu+1-\frac{1}{\alpha}}\left(1 + (\mu - \frac{1}{\alpha})e^{-\frac{nc}{k^\alpha}}\right) \frac{\left(\frac{nc}{k^\alpha}\right)^{\mu-\frac{1}{\alpha}}}{\mu-\frac{1}{\alpha}}$ which leads to $\gamma\left(\mu - \frac{1}{\alpha}, \frac{nc}{k^\alpha}\right) = \mathcal{O}(\mu^{-1}n^{-\frac{1}{\alpha}})$ when $k > n^{\frac{1}{\alpha-1}}$. Lemma 19 implies that $\Gamma\left(\mu - \frac{1}{\alpha}, nc\right) \leq B(nc)^{\mu-\frac{1}{\alpha}}e^{-nc}$ for some constant $B$ and for every $1 < \mu < n^{\frac{1}{\alpha+1}}$. This and the recursion $\Gamma(s+1, t) = s\Gamma(s, x) + x^s e^{-x}$ lead to $\Gamma\left(\mu - \frac{1}{\alpha}, nc\right) = \mathcal{O}(\frac{1}{n})$ for $1 \leq \mu < n^{\frac{1}{\alpha+1}}$ and therefore $(d)$. Also, $(a)$ is followed by the integration bound for a uni-modal series, $(b)$ is by changing of variables $\frac{nc}{x^\alpha} = y$, and $(c)$ is by the definition of Gamma function and the fact that $e^{-t}t^\mu$ is maximized at $t = \mu$ followed by Stirling's approximation. $\qquad\square$

**Lemma 16.** *For a power-law distribution with power $\alpha$ and $\mu < n^{\frac{1}{\alpha+1}}$,*

$$
\Pr\left[\Phi_\mu < \frac{\mathbb{E}[\Phi_\mu]}{2}\right] \leq \exp\left(-\frac{1}{6\mu}\left(\frac{n}{\mu}\right)^{\frac{1}{\alpha}}\right)
$$

*Proof.* $\Phi_\mu = \sum_x \mathbb{1}_x^\mu$, and therefore is a sum of independent random variables $\mathbb{1}_x^\mu$. By Bernstein's inequality

$$
\Pr\left[\left|\Phi_\mu - \mathbb{E}[\Phi_\mu]\right| > t\right] \leq 2\exp\left(-\frac{t^2/2}{\text{Var}(\Phi_\mu) + t/3}\right)
$$

Substituting $\mathbb{E}[\Phi_\mu]$ from Lemma 15 and using $\text{Var}(\Phi_\mu) \leq \mathbb{E}[\Phi_\mu]$, for $t = \frac{\mathbb{E}[\Phi_\mu]}{2}$ we have the lemma. $\qquad\square$

**Lemma 17** ( [OD12]). *Let $D$ be the number of distinct categories and $d = \mathbb{E}[D]$. Also let $v = \mathbb{E}[\Phi_1]$ be the expected number of categories that appeared once. Then,*

$$
\Pr[D < d - \sqrt{2vs}] \leq e^{-s}
$$

**Lemma 18.** *Let $X \sim POI(x)$, then for $x_0 > 0$, $\Pr(X \geq x + x_0) \leq e^{x_0 - (x+x_0)\ln(1+\frac{x_0}{x})}$, and Also $\Pr(X \leq x - x_0) \leq e^{x_0 - (x+x_0)\ln(1+\frac{x_0}{x})}$.*

*Proof.* Chernoff bound suggests that for every $t > 0$

$$\Pr(X \geq a) \leq \frac{\mathbb{E}[e^{tX}]}{e^{t \cdot a}},$$

and similarly for every $t < 0$,

$$\Pr(X \leq a) \leq \frac{\mathbb{E}[e^{tX}]}{e^{t \cdot a}}.$$

Moment generating function, $\mathbb{E}[e^{tX}]$ for $X$ distributed according to $\text{POI}(x)$ is $e^{x(e^t - 1)}$. Therefore,

$$\begin{aligned}
\Pr(X \geq a) &\leq \inf_{t > 0} \frac{e^{x(e^t - 1)}}{e^{t \cdot a}} \\
&= \inf_{t > 0} e^{x(e^t - 1) - t \cdot a} \\
&= e^{a - x - a \ln \frac{a}{x}}.
\end{aligned}$$

Substituting $a$ by $x + x_0$ leads to the lemma. $\qquad\square$

**Lemma 19** ( [NP00]). *For $a > 1$, $B > 1$, and $x > \frac{B}{B-1}(a-1)$, we have*

$$x^{a-1}e^{-x} < |\Gamma(a, x)| < B x^{a-1} e^{-x}.$$

**Lemma 20** (Theorem 4.1 in [Neu13]). *For $a > 0$ and $x > 0$, we have*

$$\exp\left(-\frac{ax}{a+1}\right) \leq \frac{a}{x^a}\gamma(a, x) \leq \frac{1}{a+1}(1 + ae^{-x}).$$

**Lemma 21.** *For every distribution, $\mu \geq 1$, and in the presence of Poisson sampling,*

$$\text{Var}(\Phi_\mu) \leq \mathbb{E}[\Phi_\mu], \quad \text{Var}(S_\mu) \leq \frac{(\mu+1)(\mu+2)}{n^2}\mathbb{E}[\Phi_{\mu+2}], \quad \mathbb{E}[S_\mu] = \frac{\mu+1}{n}\mathbb{E}[\Phi_{\mu+1}]$$

*Proof.* We use the property of the Poisson sampling that the counts are independent. For the variance of $\Phi_\mu$ we can write:

$$\begin{aligned}
\text{Var}(\Phi_\mu) &= \text{Var}\left(\sum_j \mathbb{1}_j^\mu\right) \\
&= \sum_j \text{Var}(\mathbb{1}_j^\mu) \\
&\leq \sum_j \mathbb{E}[\mathbb{1}_j^\mu] \\
&= \mathbb{E}[\Phi_\mu].
\end{aligned}$$

Also, for the expected value of the sum of probabilities that appeared $\mu$ times, we have:

$$\begin{aligned}
\mathbb{E}[S_\mu] &= \mathbb{E}\left[\sum_j p_j \mathbb{1}_j^\mu\right] \\
&= \sum_j p_j e^{-np_j} \frac{(np_j)^\mu}{\mu!} \\
&= \frac{\mu+1}{n} \sum_j e^{-np_j} \frac{(np_j)^{\mu+1}}{(\mu+1)!} \\
&= \frac{\mu+1}{n} \sum_j \mathbb{E}[\mathbb{1}_j^{\mu+1}] \\
&= \frac{\mu+1}{n} \mathbb{E}[\Phi_{\mu+1}],
\end{aligned}$$

and for their variance, we can write:

$$\mathrm{Var}[S_\mu] = \mathrm{Var}\Big[\sum_j p_j \mathbb{1}_j^\mu\Big]$$

$$= \sum_j p_j^2 \mathrm{Var}(\mathbb{1}_j^\mu)$$

$$= \sum_j p_j^2 \mathbb{E}[\mathbb{1}_j^\mu]$$

$$= \sum_j p_j^2 e^{-np_j} \frac{(np_j)^\mu}{\mu!}$$

$$= \frac{(\mu+1)(\mu+2)}{n^2} \sum_j e^{-np_j} \frac{(np_j)^{\mu+2}}{(\mu+2)!}$$

$$= \frac{(\mu+1)(\mu+2)}{n^2} \sum_j \mathbb{E}[\mathbb{1}_j^{\mu+2}]$$

$$= \frac{(\mu+1)(\mu+2)}{n} \mathbb{E}[\Phi_{\mu+2}].$$

$\square$

**Lemma 22.** *Let $\Phi_1$ be number of categories appeared once. Also let $D$ be the number of distinct categories observed and $d = \mathbb{E}[D]$, then for $0 < t < 1$,*

$$\Pr\left(\left|\frac{\Phi_1}{D} - \frac{\mathbb{E}[\Phi_1]}{d}\right| > \frac{2t}{1-t}\right) \leq 4\exp\left(-\frac{t^2 \mathbb{E}[\Phi_1]}{2(1+t/3)}\right)$$

*Proof.* Using Lemma 16 we have

$$\Pr\left(\left|\frac{\Phi_1}{\mathbb{E}[\Phi_1]} - 1\right| > t\right) = \Pr\left(\left|\Phi_1 - \mathbb{E}[\Phi_1]\right| > t\mathbb{E}[\Phi_1]\right)$$

$$\leq \exp\left(-\frac{t^2 \mathbb{E}[\Phi_1]}{2(1+t/3)}\right)$$

Similarly for number of distinct elements we have

$$\Pr\left(\left|\frac{D}{d} - 1\right| > t\right) = \Pr\left(|D - d| > td\right)$$

$$\overset{(a)}{\leq} \Pr\left(|D - d| > t\mathbb{E}[\Phi_1]\right)$$

$$\overset{(b)}{\leq} \exp\left(-\frac{t^2 \mathbb{E}[\Phi_1]}{2(1+t/3)}\right)$$

where $(a)$ is because $\mathbb{E}[\Phi_1] \leq d$ and $(b)$ is because $\mathrm{Var}(D) = \mathbb{E}[\Phi_1]$. Hence, with probability at least $1 - 4\exp\left(-\frac{t^2 \mathbb{E}[\Phi_1]}{2(1+t/3)}\right)$ we have $1-t \leq \frac{\Phi_1}{\mathbb{E}[\Phi_1]} \leq 1+t$ and $1-t \leq \frac{D}{d} \leq 1+t$. With probability $\geq 1 - 4\exp\left(-\frac{t^2 \mathbb{E}[\Phi_1]}{2(1+t/3)}\right)$, $\frac{1-t}{1+t} \leq \frac{\Phi_1}{D}\frac{d}{\mathbb{E}[\Phi_1]} \leq \frac{1+t}{1-t}$, namely $\left|\frac{\Phi_1}{D} - \frac{\mathbb{E}[\Phi_1]}{d}\right| \leq \max\left(\frac{1+t}{1-t} - 1, 1 - \frac{1-t}{1+t}\right) = \frac{2t}{1-t}$. $\square$