[Reviews · NeurIPS 2017]

Reviewer 1



This paper presents a theoretical examination of the optimality of absolute discounting similar to the examination of optimality of Good Turing in Orlitsky and Suresh (2015). Results for minimax optimality, adaptivity and competitiveness are presented, as well as an equivalence between absolute discounting and Good Turing in certain scenarios, which suggests a choice of discount. Experimental results demonstrate the quality of the approach, along with some interesting results on predicting terror attacks in classes of cities (e.g., cities with zero prior attacks) given prior data. The paper is very well written and crystal clear, and the results are quite interesting. This is an excellent addition to the literature on these methods. I have two minor quibbles with how absolute discounting is presented. First, it is stated in the introduction and after equation 1 that in the NLP community absolute discounting has long been recognized as being better than Good Turing for sequence models. That's not entirely accurate. What has been recognized is that Kneser-Ney smoothing, which imposes marginal distribution constraints on absolute discounted language models, is superior. Absent the marginal distribution constraints of the Kneser Ney method, absolute discounting is rarely chosen over Katz smoothing (based on Good Turing). Due to the simplicity of absolute discounting, the imposition of these constraints becomes an easy closed-form update, hence its use as the basis for that technique. If such constraints were easy to apply to Good-Turing then perhaps that would have been the basis of the approach. None of this negates anything that you argue for in your paper, but you should just be a bit clearer about absolute discounting being superior to Good Turing. Second, your presentation of absolute discounting in equation 1 really does imply that delta is between 0 and 1, even though that is not stated until later. If you want to have a more general discounting presentation which allows discounts greater than 1 (which is often how it is presented in the literature using a max with the resulting value and 0) then you'll need to account for the case where delta is greater than the count. As it stands, the presentation is quite clean, so you may want to just be explicit that the presentation assumes the discount is less than 1, which is the interesting scenario, and explain in a footnote what kinds of modifications would be required otherwise. I also think you could give an easy intuition about the formula for the case when mu_j = 0 in equation 1 in a single sentence by pointing out that (1) there is D*delta discounted mass (numerator) which (2) must be shared among the k-D items with zero counts and normalized (denominator). Obvious, yes, once you work through it, but probably could save the reader a bit of time with such an intuition. Otherwise, I thoroughly enjoyed the paper and think it is a nice addition to the literature.

Reviewer 2



SUMMARY: The paper discusses methods for estimating categorical distributions, in particular a method called "absolute discounting". The paper gives bounds for the KL-risk of absolute discounting, and discusses properties such as minimax rate-optimality, adaptivity, and its relation to the Good–Turing estimator. CLARITY: The paper is presented in reasonably clear language, and is well-structured. NOVELTY: The contributions of the paper seem good, but perhaps rather incremental, and not ground-breakingly novel. The paper advocates for the use of absolute discounting, and gives good arguments in favor, including theoretical properties and some experimental results. But the technique as such isn't especially novel, and more general versions of it exist that aren't referenced. The literature review might not be thorough enough: for example, there are many relevant techniques in Chen & Goodman's comprehensive and widely-cited report (1996, 1998) that aren't mentioned or compared to. The paper does cite this report, but could perhaps engage with it better. Also, the paper does not give any insight on its relation to clearly relevant Bayesian techniques (e.g. Pitman-Yor processes and CRPs) that are similar to (and perhaps more general than) the form absolute discounting presented in this paper. These other techniques probably deserve a theoretical and/or experimental comparison. At the very least they could be mentioned, and it would be helpful to have their relationship to "absolute discounting" clarified. SCORE: Overall, I enjoyed reading the paper, and I hope the authors will feel encouraged to continue and improve their work. The main contributions seem useful, but I have a feeling that their relevance to other techniques and existing research isn't clarified enough in this version of the paper. A few minor suggestions for improvements to the paper follow below. SECTION 8 (Experiments): Some questions. Q1: Is 500 Monte-Carlo iterations enough? Q2: If the discount value is set based on the data, is that cheating by using the data twice? LANGUAGE: Some of the phrasing might be a bit overconfident, e.g.: Line 1: "Categorical models are the natural fit..." -> "Categorical models are a natural fit..."? Line 48: "[we report on some experiments], which showcases perfectly the all-dimensional learning power of absolute discounting" (really, "perfectly"?) Line 291: "This perfectly captures the importance of using structure [...]" NOTATION: Totally minor, but perhaps worth mentioning as it occurs a lot: "Good-Turing" should be written with an en-dash ("–" in Unicode, "--" in LaTeX), and not with a hyphen ("-" in Unicode, "-" in LaTeX). BIBLIOGRAPHY: In the bibliography, "Good–Turing" is often written in lowercase ("good-turing"), perhaps as a result of forgetting the curly brace protections in the BibTeX source file. (Correct syntax: "{G}ood--{T}uring".) The final reference (Zip35, Line 364) seems incomplete: only author, title and year are mentioned, no information is given where or how the paper was published.

Reviewer 3



Summary: The authors study the performance of absolute discounting estimator of discrete distributions from iid samples. The competitive loss performance (introduced in last year's NIPS best paper) is used primarily as the metric. The main result is to show strong guarantees of the absolute discounting method (widely used in NLP including in KN smoothing) in a large variety of adaptive distribution families (including the power law distributions). The connection to Good-Turing performance is especially striking. Comments: This is an outstanding paper worthy of an oral slot. It continues the line of research introduced in [OS16]: "Competitive Distribution Estimation: Why is Good-Turing Good" and especially the competitive optimality setting for a large family of distributions. The main results are aptly summarized in lines 141-163, where one sees that absolute discounting inherits much of the same properties as Good Turing, especially in the context of power law distribution families. I have no comments on the proofs -- they are very nicely done, of which I found the proof Theorem 6 to be the most involved and creative. One question I have is regarding the following: in a then-seminal work Yee Whye Teh showed that the hierarchical Chinese restaurant process parameters can be naturally learnt by an absolute discounting method -- this provided a Bayesian context to KN smoothing and modified KN smoothing techniques in language modeling. The standard reference is @inproceedings{Teh2006AHB, title={A Hierarchical Bayesian Language Model Based On Pitman-Yor Processes}, author={Yee Whye Teh}, booktitle={ACL}, year={2006} } but various other summaries and (even a NIPS tutorial) are available online. How does the present work relate to this body of work? Is there a sense in which the hierarchical Pitman Yor process represent the argument to the minimax settings of this paper?